# ReST-KV: Robust KV Cache Eviction with Layer-wise Output Reconstruction and Spatial-Temporal Smoothing

**Yongqi An**[1,2], **Chang Lu**[3], **Kuan Zhu**[1,2], **Tao Yu**[1,2], **Chaoyang Zhao**[1,2],
**Hong Wu**[3], **Ming Tang**[1,2], **Jinqiao Wang**[1,2,4,5*]

[1]Foundation Model Research Center, Institute of Automation, Chinese Academy of Sciences, Beijing, China
[2]School of Artificial Intelligence, University of Chinese Academy of Sciences, Beijing, China
[3]University of Electronic Science and Technology of China, Chengdu, China
[4]Wuhan AI Research, Wuhan, China      [5]Objecteye Inc., Beijing, China
yongqi.an@nlpr.ia.ac.cn, jqwang@nlpr.ia.ac.cn

## Abstract

Large language models (LLMs) face growing challenges in efficient generative inference due to the increasing memory demands of Key-Value (KV) caches, especially for long sequences. Existing eviction methods typically retain KV pairs with high attention weights but overlook the impact of attention redistribution caused by token removal, as well as the spatial-temporal dynamics in KV selection. In this paper, we propose **ReST-KV**, a robust KV eviction method that combines layer-wise output **Re**construction and **S**patial-**T**emporal smoothing to provide a more comprehensive perspective for the KV cache eviction task. Specifically, ReST-KV formulates KV cache eviction as an optimization problem that minimizes output discrepancies through efficient layer-wise reconstruction. By directly modeling how each tokens removal affects the model output, our method naturally captures attention redistribution effects, going beyond simplistic reliance on raw attention weights. To further enhance robustness, we design exponential moving average smoothing to handle temporal variations and an adaptive window-based mechanism to capture spatial patterns. Our method, ReST-KV, significantly advances performance on long-context benchmarks. It surpasses state-of-the-art baselines by 2.58% on LongBench and 15.2% on RULER. Additionally, ReST-KV consistently outperforms existing methods on Needle-in-a-Haystack and InfiniteBench, all while achieving a remarkable $10.61\times$ reduction in decoding latency at 128k context length. The code is included in the supplementary material and is designed for easy reproduction.

## 1 Introduction

Large language models (LLMs)(Achiam et al., 2023; Anthropic, 2023; Dubey et al., 2024; MistralAI, 2023) have significantly advanced natural language processing (NLP). These models have enabled breakthroughs in various tasks, such as document summarization(Zhang et al., 2024a), multi-turn dialogues (Du et al., 2021), retrieval augmentation (Yao et al., 2022), and code generation (Roziere et al., 2023). Recent models like GPT-4 (Achiam et al., 2023), Claude 3.5 (Anthropic, 2023), and Llama-3.1 (Dubey et al., 2024) have extended their context lengths beyond 128K tokens, allowing for long-context applications. However, as context length increases, the memory required to store KV cache grows rapidly, potentially reaching hundreds of gigabytes when handling longer sequences. Thus, optimizing KV cache during inference, without retraining, is crucial for improving both efficiency and scalability.

KV cache eviction, which identifies and removes less important KV pairs, is a promising approach to reduce memory consumption and enhance computational efficiency (Li et al., 2024a). Current methods typically rely on fixed attention patterns (Han et al., 2024; Ge et al., 2023) or use statistical

---

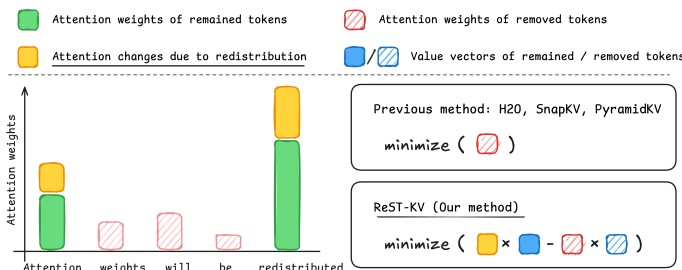

Figure 1: Comparison between ReST-KV and existing methods. Unlike prior approaches that overlook attention redistribution, ReST-KV considers its impact to improve KV retention.

information from attention weights (Zhang et al., 2023; Li et al., 2024b; Cai et al., 2024) to estimate the importance of KV pairs. However, as shown in Figure 1, these approaches focus solely on retaining query-key pairs with high similarity scores, while ignoring the attention redistribution effects caused by removing certain pairs. This redistribution can alter the overall attention landscape, leading to suboptimal retention decisions and degraded performance, especially under tight cache constraints.

In this paper, we propose ReST-KV, a robust KV cache eviction method that accounts for the effects of attention redistribution and the spatial-temporal dynamics in KV selection. We revisit the KV cache eviction problem and reformulate it as preserving the attention output at each layer under fixed memory constraints. Specifically, we measure the reconstruction loss caused by removing each individual KV pair, and use it as an eviction indicator: the larger the loss, the more important the KV pair. This loss implicitly captures the impact of attention redistribution caused by the removal. Moreover, our empirical observations show that KV importance varies significantly across both time and space. To further improve robustness, we introduce two smoothing mechanisms: (1) an exponential moving average to model temporal dynamics by emphasizing more recent KV pairs, and (2) an adaptive window-based spatial smoothing method, which adjusts for varying window sizes and offsets by estimating the spatial dynamics.

By evaluating on a wide range of downstream tasks including LongBench, RULER, Needle-in-a-Haystack, and InfiniteBench, we demonstrate that ReST-KV consistently outperforms state-of-the-art baselines, especially under low cache budgets and demonstrates more robustness in multi-turn dialogue scenarios. We extensively evaluate ReST-KV on challenging long-context benchmarks such as LongBench, RULER, Needle-in-a-Haystack, and InfiniteBench. Our results show it consistently surpasses state-of-the-art baselines, with particularly strong gains of 2.58% on LongBench and 15.2% on RULER. ReST-KV also exhibits greater robustness in multi-turn dialogue and efficiency under constrained cache budgets. For decoding, it achieves a $10.61\times$ latency reduction at 128k context length when integrated with FlashAttention-2. Importantly, ReST-KV is fully compatible with existing prefill sparse attention methods, leading to a $2.37\times$ TTFT speedup. In summary, we make the following contributions:

- A novel formulation of KV eviction treating it as layer-wise output reconstruction, enabling a new importance indicator that captures attention redistribution effects.
- A spatial-temporal smoothing mechanism combining exponential moving average and adaptive windowing, significantly enhancing robustness in KV selection.
- Extensive experiments show that ReST-KV outperforms state-of-the-art baselines under low cache budgets and reduces decoding latency by up to $10\times$ at a 128k context length.

## 2 RELATED WORK

### 2.1 KV CACHE EVICTION

KV cache eviction, a prominent method for optimizing KV cache during inference without retraining, alleviates memory and latency issues in long-context LLMs (Li et al., 2024a). Early eviction methods focused on specific attention patterns, such as StreamingLLM (Xiao et al., 2023) and LM-Infinite (Han et al., 2024), retain only the initial and local tokens. While more flexible approaches like FastGen (Ge et al., 2023) and RazorAttention (Tang et al., 2024) were developed, they still rely on predefined patterns and risk ignoring important tokens. Subsequent studies introduced eviction

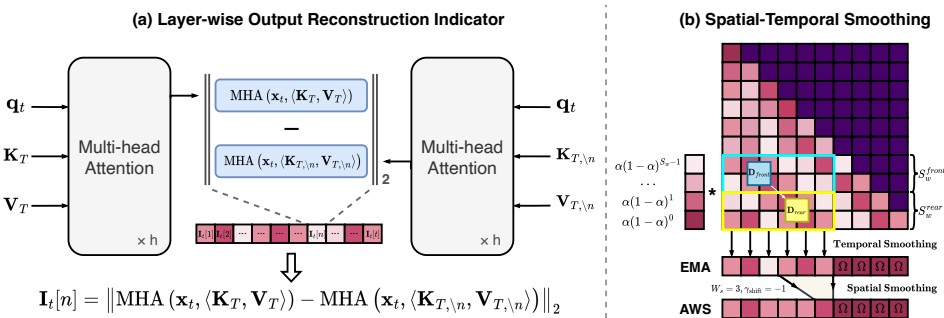

Figure 2: Overview of ReST-KV. (a) Layer-wise output reconstruction quantifies each KV pairs impact on output error as its eviction indicator. (b) Two smoothing mechanisms enhance robustness: exponential moving average for temporal smoothing and an adaptive window-based approach for spatial smoothing.

indicators to assess the importance of KV cache entries, often using attention weights. For instance, H2O (Zhang et al., 2023) uses cumulative attention weights, and SnapKV (Li et al., 2024b) pools the average attention weight over the last window. In addition to indicator improvements, some research has explored non-uniform layer-wise and head-wise budget allocation strategies. PyramidKV (Cai et al., 2024) and PyramidInfer (Yang et al., 2024) allocate budget in a pyramid fashion, while DynamicKV (Zhou et al., 2024), D2O (Wan et al., 2024) and CAKE (Qin et al., 2025) adaptively allocate budget based on layer-specific information. AdaKV (Feng et al., 2024) adjusts the budget per head based on output $\ell_1$ loss bounds. Our work focuses on the limitations of existing eviction indicators, which primarily rely on attention weights derived from query-key interactions and overlook the combined impact of value vectors and spatial-temporal dynamics. Furthermore, our approach is fully compatible with existing layer-wise and head-wise budget allocation strategies.

## 2.2 ATTENTION DYNAMICS

While attention is central to the success of Transformers, it also poses scalability challenges in long-context settings due to its quadratic complexity. Recent work has therefore investigated attention dynamicsspecifically, the spatiotemporal patterns and redistribution of attention weightsas a means to enable more efficient inference.

Several studies reveal structured attention behaviors. MInference (Jiang et al., 2024) discovers a "vertical-slash" pattern, where attention gradually shifts across tokens over time, indicating evolving token importance. FlexPrefill (Lai et al., 2025) similarly identifies consistent attention trajectories during prefill. Keyformer (Adnan et al., 2024) examines how KV eviction distorts attention distributions and proposes normalization to mitigate such shifts.

Distinct from the above methods, we reformulate KV cache eviction by explicitly modeling attention redistribution and spatiotemporal dynamics. Rather than relying solely on static attention weights, our approach captures temporal evolution and layer-wise shifts in attention, enabling more robust importance estimation and significantly improving performance under memory constraints.

## 3 METHODOLOGY

### 3.1 PRELIMINARY

LLMs typically decode text in an auto-regressive manner, which allows them to generate high-quality, contextually coherent text. However, this decoding process is computationally expensive, as it involves a high degree of repetitive calculations, making it challenging to apply in real-time or large-scale scenarios.

KV cache, a widely recognized technique, reduces redundant computation by storing previously computed keys and values. In this section, we describe the attention computation under the KV cache framework, laying the foundation for our discussion on KV cache eviction. For clarity, we focus on a single attention head and layer, omitting footnotes. At each decoding step $t$, the KV cache stores previously computed keys and values $\langle \mathbf{K}_{1:t-1}, \mathbf{V}_{1:t-1} \rangle$ for $X[1:t-1]$, enabling reuse in future steps. For convenience, we denote $\mathbf{K}_{1:t-1}$ as $\mathbf{K}_{T-1}$ and $\mathbf{V}_{1:t-1}$ as $\mathbf{V}_{T-1}$. Consequently, the model

requires only the current token $\mathbf{x}_t$ to generate $\mathbf{x}_{t+1}$, rather than the full sequence $X = [\mathbf{x}_1, \ldots, \mathbf{x}_t]$. Formally, at step $t$, the query $\mathbf{q}_t$, key $\mathbf{k}_t$, and value $\mathbf{v}_t$ are computed as:

$$\mathbf{q}_t = \mathbf{x}_t \mathbf{W}_Q, \ \mathbf{k}_t = \mathbf{x}_t \mathbf{W}_K, \ \mathbf{v}_t = \mathbf{x}_t \mathbf{W}_V, \tag{1}$$

where $\mathbf{W}_Q, \mathbf{W}_K, \mathbf{W}_V$ are the components of the $\mathbf{Q}, \mathbf{K}, \mathbf{V}$ weight matrices corresponding to a single attention head. The currently computed $\mathbf{k}_t$ and $\mathbf{v}_t$ will be concatenated with the previously cached keys and values, and used in the attention computation for decoding step $t$:

$$\mathbf{K}_T = \text{Concat}\left(\mathbf{K}_{T-1}, \mathbf{k}_t\right), \mathbf{V}_T = \text{Concat}\left(\mathbf{V}_{T-1}, \mathbf{v}_t\right), \tag{2}$$

where $\mathbf{K}_T$ and $\mathbf{V}_T$ are the entire sequences of keys and values at decoding step $t$. The attention output $\mathbf{z}_t$ for the token $\mathbf{x}_t$ at step $t$ is calculated as:

$$\mathbf{z}_t = \text{softmax}\left(\frac{\mathbf{q}_t \mathbf{K}_T^\top}{\sqrt{d_k}}\right) \mathbf{V}_T = \mathbf{A}_t \mathbf{V}_T, \tag{3}$$

where $\mathbf{A}_t$ represents the attention weights for the token $\mathbf{x}_t$ and is used by existing methods to compute eviction indicators. $d_k$ represents the dimension of the key vectors in the attention mechanism.

Finally, the output of a single head in the multi-head attention can be expressed as:

$$\text{MHA}\left(\mathbf{x}_t, \langle \mathbf{K}_T, \mathbf{V}_T \rangle\right) = \mathbf{z}_t \mathbf{W}_O, \tag{4}$$

where $\mathbf{W}_O$ is the weight matrix of output projection corresponding to a single attention head.

## 3.2 LAYER-WISE RECONSTRUCTION INDICATOR

We reformulate KV cache eviction as preserving the attention output distribution at each layer under fixed memory constraints, naturally capturing the effects of attention redistribution. We formalize this paradigm as *layer-wise reconstruction*, a framework that aligns with the transformer's inherent layer-wise computation flow. Specifically, for a single layer, the subproblem is expressed as:

**Definition 3.1.** Given a cache budget $B$ for a single layer, the task is to select a series of important KV cache entries $\langle \hat{\mathbf{K}}_T, \hat{\mathbf{V}}_T \rangle$ containing up to $B$ elements from the total cache entries $\langle \mathbf{K}_T, \mathbf{V}_T \rangle$ at the step $t$, with the goal of maximizing the retention of the orignial MHA output. We use $\ell_2$ distance to calculate reconstruction error, the objective for a single attention head can be defined as:

$$\underset{\langle \hat{\mathbf{K}}_T, \hat{\mathbf{V}}_T \rangle}{\arg\min} \ \left\| \text{MHA}\left(\mathbf{x}_t, \langle \mathbf{K}_T, \mathbf{V}_T \rangle\right) - \text{MHA}\left(\mathbf{x}_t, \langle \hat{\mathbf{K}}_T, \hat{\mathbf{V}}_T \rangle\right) \right\|_2$$

$$\text{s.t.} \quad \left| \langle \hat{\mathbf{K}}_T, \hat{\mathbf{V}}_T \rangle \right| \leq B, \tag{5}$$

where $\left| \langle \hat{\mathbf{K}}_T, \hat{\mathbf{V}}_T \rangle \right|$ is the number of selected KV pairs.

To efficiently compute Eq.5, we adopt a greedy selection strategy that retains the top-$B$ KV pairs estimated to have the greatest impact on the attention output. Specifically, for the $n$-th KV pair, its importance is measured by the increase in reconstruction error when it is removed, which based on the local linearity assumptions (Molchanov et al., 2016). The eviction indicator is defined as:

$$\mathbf{I}_t[n] = \left\| \text{MHA}(\mathbf{x}_t, \langle \mathbf{K}_T, \mathbf{V}_T \rangle) \right.$$
$$\left. - \text{MHA}(\mathbf{x}_t, \langle \mathbf{K}_{T,\backslash n}, \mathbf{V}_{T,\backslash n} \rangle) \right\|_2, \tag{6}$$

where $\langle \mathbf{K}_{T,\backslash n}, \mathbf{V}_{T,\backslash n} \rangle$ represents the set of cache with the $n$-th KV pair removed.

By introducing Eq. 3 and Eq. 4 for derivation, Eq. 6 can be simplified as follows:

$$\mathbf{I}_t[n] = \frac{\mathbf{A}_t[n]}{1 - \mathbf{A}_t[n]} \left\| \text{MHA}\left(\mathbf{x}_t, \langle \mathbf{K}_T, \mathbf{V}_T \rangle\right) - \mathbf{v}_n \mathbf{W}_O \right\|_2, \tag{7}$$

where $\mathbf{A}_t[n]$ represents the attention weights of the query $\mathbf{q}_t$ with respect to the key $\mathbf{k}_n$, and $\mathbf{v}_n$ represents the $n$-th value in the value cache $\mathbf{V}_T$.

Traditional eviction indicators only considered $\mathbf{A}_t[n]$, neglecting the effects of attention redistribution. Eq. 7 demonstrates that the importance of a KV pair depends on two mechanisms:

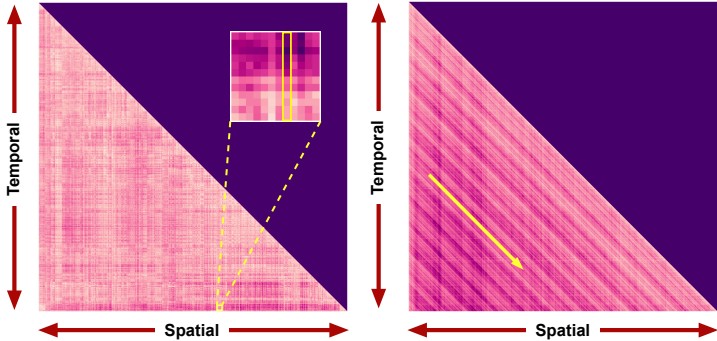

Figure 3: Visualization analysis of the spatial-temporal dynamics of the output reconstruction indicator. The left plot shows dynamic temporal variations in KV pair importance over steps, with the zoomed-in view highlighting a KV pairs gradual decline in importance. The right plot reveals spatial shifts, where similar importance patterns emerge at shifted positions.

- **Nonlinear Attention Reweighting**: The first term $\frac{\mathbf{A}_t[n]}{1-\mathbf{A}_t[n]}$ acts as a monotonic nonlinear amplifier in $(0,1)$. While preserving the conventional principle that higher attention weights $\mathbf{A}_t[n]$ indicate stronger retention priority, this transformation introduces curvature to better discriminate between high-competition KV pairs compared to linear scaling in prior methods.
- **Redistribution Sensitivity**: The second term $\|\text{MHA}(\cdot) - \mathbf{v}_n \mathbf{W}_O\|_2$ captures the redistribution of attention after removing the $n$-th KV pair. It reflects how much the remaining KV pairs fail to compensate for the excluded value in reconstructing the MHA output. A smaller discrepancy indicates that attention can be effectively redistributed to preserve the output, thus signaling lower importance of the removed KV pair.

The additional analysis and the derivation of Eq. 7 can be found in Appendix A and Eq. 21.

## 3.3 SPATIAL-TEMPORAL SMOOTHING

To enhance the robustness of KV pair selection during the prefill stage, we analyze the spatial-temporal dynamics of the KV pairs' reconstruction error (Eq. 7). From Figure 3, we observe two key characteristics: (1) The importance of KV pairs exhibits dynamic temporal variations (i.e., the fluctuating patterns of $\mathbf{I}_1[n], \mathbf{I}_2[n], \ldots, \mathbf{I}_t[n]$ along the temporal dimension, and (2) simultaneously demonstrates dynamic spatial shifts where similar importance distributions emerge across shifted positions (e.g., $\mathbf{I}_{t-k}[n-kN], \ldots, \mathbf{I}_{t-1}[n-N], \mathbf{I}_t[n]$ exhibit analogous patterns).

Leveraging these observations, we introduce two novel smoothing mechanisms to enhance the robustness of KV pair selection, as illustrated in Figure 2(b). These mechanisms address temporal variations and spatial shifts in KV pair importance, ensuring a more stable and reliable selection process. By applying these techniques, we aim to reduce short-term fluctuations and capture long-term trends, ultimately improving the performance of the KV cache eviction.

**Exponential Moving Average Temporal Smoothing.** Inspired by SnapKV (Li et al., 2024b), we use a recent query window $S_w$ to assess the importance of KV pairs. To model temporal dynamics, we apply exponential moving average (EMA) smoothing to the importance of KV pairs, which assigns higher weights to recent queries while dampening earlier fluctuations. To apply this smoothing over a limited window of recent queries, we define the temporal smoothing as:

$$\hat{\mathbf{I}}_t[n] = \begin{cases} \text{EMA}(\mathbf{I}_{t-S_w:t}[n]), & \text{if } n < t - S_w, \\ \Omega, & \text{otherwise,} \end{cases} \quad (8)$$

where $\hat{\mathbf{I}}_t[n]$ represents the eviction indicator with temporal smoothing. $\text{EMA}(\cdot)$ captures the temporal variation in importance. We assign an arbitrarily large value $\Omega$ to the most recent $S_w$ tokens to ensure their preservation.

The exponential moving average $\text{EMA}(\cdot)$ is defined as:

$$\text{EMA}(\mathbf{I}_{t_1:t_2}[n]) = \begin{cases} \alpha \mathbf{I}_{t_2}[n] + (1-\alpha)\,\text{EMA}(\mathbf{I}_{t_1:t_2-1}[n]), \\ \qquad\qquad\qquad\qquad \text{if} \quad t_1 < t_2, \\ \mathbf{I}_{t_1}[n], \qquad\qquad\quad \text{elif} \quad t_1 = t_2, \end{cases} \quad (9)$$

where $\mathrm{EMA}(\mathbf{I}_{t_1:t_2}[n])$ represents the exponential moving average of the reconstruction errors $\mathbf{I}_{t_1}[n], \dots, \mathbf{I}_{t_2}[n]$ computed over the steps from $t_1$ to $t_2$. $\alpha$ is the smoothing factor that controls the weight of the current reconstruction error $\mathbf{I}_{t_2}[n]$ relative to the previous error $\mathrm{EMA}(\mathbf{I}_{t_1:t_2-1}[n])$ in the update process.

**Adaptive Window-Based Spatial Smoothing.** To capture spatial shifts in KV importance over time, we split the observation window into two halves: $S_w^{\mathrm{front}}$ and $S_w^{\mathrm{rear}}$. For each half, we compute the average index of the top-$B$ important KV pairs:

$$\mathbf{D}_{\mathrm{front}} = \frac{2}{B \cdot S_w} \sum_{t \in S_w^{\mathrm{front}}} \sum_B \underset{B}{\mathrm{argmax}} \left( \mathbf{I}_t \right), \tag{10}$$

where $\frac{2}{B \cdot S_w}$ is a normalization factor. $S_w^{\mathrm{front}}$ denotes the first half of queries within the input window $S_w$. $\mathbf{D}_{\mathrm{rear}}$ is computed similarly for the second half of the queries. The difference $\Delta D = \mathbf{D}_{\mathrm{rear}} - \mathbf{D}_{\mathrm{front}}$ reflects how KV importance shifts across positions. We use this signal to adaptively adjust both the window size and shift:

$$W_s = 2 \cdot \left\lfloor \frac{|\mathbf{D}_{\mathrm{rear}} - \mathbf{D}_{\mathrm{front}}|}{\beta} \right\rfloor + 1, \tag{11}$$

$$\gamma_{\mathrm{shift}} = \begin{cases} \lfloor \frac{\mathbf{D}_{\mathrm{front}} - \mathbf{D}_{\mathrm{rear}}}{\beta} \rfloor, & \text{if } \mathbf{D}_{\mathrm{front}} - \mathbf{D}_{\mathrm{rear}} > 0, \\ \lfloor \frac{\mathbf{D}_{\mathrm{front}} - \mathbf{D}_{\mathrm{rear}}}{\beta} \rfloor + 1, & \text{if } \mathbf{D}_{\mathrm{front}} - \mathbf{D}_{\mathrm{rear}} \le 0, \end{cases} \tag{12}$$

where $W_s$ is the window size and $\gamma_{\mathrm{shift}}$ is the shift of the sliding window. $\beta$ is a scaling factor that determines the granularity of the sliding window's movement, controlling the size of the steps taken when calculating the window shift and size. $\lfloor \cdot \rfloor$ represents the floor function, which rounds a number down to the nearest integer.

In summary, the final eviction indicator, which incorporates both layer-wise output reconstruction and spatial-temporal smoothing, is as follows:

$$\mathcal{I}_t[n] = \frac{\sum_{k=-\lfloor W_s/2 \rfloor + \gamma_{\mathrm{shift}}}^{\lfloor W_s/2 \rfloor + \gamma_{\mathrm{shift}}} \hat{\mathbf{I}}_t[k]}{W_s}. \tag{13}$$

The selected $\langle \hat{\mathbf{K}}_T, \hat{\mathbf{V}}_T \rangle$ is the subset of the original KV pairs, defined as:

$$\hat{\mathbf{K}}_T = \mathbf{K}_T[\mathbf{D}_t, :], \ \hat{\mathbf{V}}_T = \mathbf{V}_T[\mathbf{D}_t, :], \ \mathbf{D}_t = \underset{B}{\mathrm{argmax}} \left( \mathcal{I}_t \right), \tag{14}$$

where $\mathbf{D}_t$ denotes the indices of the top $B$ KV pairs based on the eviction indicator $\mathcal{I}_t$. The same operation is applied to each head and layer, and different KV pairs can be selected for different heads in each layer.

## 4 EXPERIMENTS

### 4.1 EXPERIMENTAL SETTINGS

**Backbone LLMs.** We evaluate ReST-KV on five open-source LLMs spanning two mainstream attention architectures: (1) **Multi-head attention**, Llama2-Chat (Touvron et al., 2023) and Gemma-Instruct (Team et al., 2024); (2) **Grouped-query attention**, Llama3-Instruct (Dubey et al., 2024), Mistral-Instruct-v0.3 (Jiang et al., 2023), and Qwen2.5-Instruct (Team, 2024).

**Baseline Methods.** We compare ReST-KV with five baselines: (1) Fixed Attention Patterns: StreamingLLM (Xiao et al., 2023); (2) Eviction Indicator: H2O (Zhang et al., 2023), TOVA (Oren et al., 2024), SnapKV (Li et al., 2024b), LaCache (Shi et al., 2025). We also incorporate adaptive budget strategies from PyramidKV (Cai et al., 2024) and AdaKV (Feng et al., 2024) into our method to show compatibility.

**Evaluating Tasks.** We evaluate ReST-KV on three prominent benchmarks: (1) LongBench (Bai et al., 2023), which tests long-context understanding across 16 datasets spanning six categories; and (2) RULER (Hsieh et al., 2024), a challenging long-context benchmark consisting of 4 categories and 13 complex tasks; (3) Needle-in-a-Haystack (Liu et al., 2024a), designed to assess the ability of models to retrieve key information from long sequences; (4) InfiniteBench (Zhang et al., 2024b), includes 10 tasks designed to test various aspects of long-context processing. Detailed results are reported in Appendix J.

Table 1: Performance comparison across 16 datasets of LongBench. The best result is highlighted in **bold**, and the second-best is underlined. ReST-KV achieves the best performance in most cases.

| Method | Single-Document QA | | | Multi-Document QA | | | Summarization | | | Few-shot Learning | | | Synthetic | | Code | | Avg. |
|---|---|---|---|---|---|---|---|---|---|---|---|---|---|---|---|---|---|
| | NrtvQA | Qasper | MF-en | HotpotQA | 2WikiMQA | Musique | GovReport | QMSum | MultiNews | TREC | TriviaQA | SAMSum | PCount | PRe | Lcc | RB-P | |
| *Llama-3.1-8B-Instruct, $B_{total} = 64L$* | | | | | | | | | | | | | | | | | |
| StreamingLLM | 7.65 | 5.08 | 14.14 | 10.93 | 12.64 | 6.86 | 16.57 | 18.93 | 16.30 | 38.50 | 83.13 | 34.65 | **9.78** | **96.28** | 54.16 | 48.21 | 29.61 |
| H2O | 12.23 | 5.12 | 15.12 | 11.51 | 10.14 | 6.23 | 17.23 | 19.51 | 16.79 | 39.15 | 81.51 | 36.12 | 8.12 | 95.12 | 51.25 | 47.12 | 29.52 |
| TOVA | 18.52 | 6.12 | 17.32 | 12.15 | 12.51 | 7.35 | 16.24 | 20.41 | 16.34 | 38.41 | 82.61 | 36.16 | 8.14 | 95.23 | 55.21 | 47.35 | 30.63 |
| SnapKV | 19.90 | 5.78 | 18.38 | 13.51 | 14.42 | 8.52 | 17.35 | 20.44 | 17.33 | 41.00 | 85.37 | 37.63 | 8.93 | 91.08 | 55.09 | **48.88** | 31.48 |
| ReST-KV | **22.43** | **7.19** | **19.25** | **14.11** | **15.04** | 7.97 | **20.56** | **21.10** | **19.15** | **53.50** | **88.23** | **40.21** | 8.46 | 93.90 | **56.74** | 48.77 | **33.54** |
| *Llama-3.1-8B-Instruct, $B_{total} = 512L$* | | | | | | | | | | | | | | | | | |
| StreamingLLM | 19.15 | 6.47 | 15.02 | 10.94 | 12.58 | 6.23 | 23.66 | 20.05 | 23.31 | 57.50 | 87.70 | 41.86 | **10.25** | 90.74 | 62.39 | 53.61 | 33.84 |
| H2O | 26.23 | 7.34 | 20.51 | 11.52 | 13.52 | 7.34 | 23.23 | 21.24 | 23.14 | 58.50 | 86.12 | 40.15 | 7.25 | 91.02 | 61.23 | 54.12 | 34.53 |
| TOVA | 27.34 | 8.34 | 22.45 | 12.25 | 14.51 | 8.42 | 24.23 | 22.13 | 22.25 | 58.50 | 89.31 | 40.51 | 8.24 | 93.14 | 62.23 | 55.61 | 35.59 |
| SnapKV | 28.02 | 9.83 | 24.84 | 13.77 | 15.40 | 10.21 | 25.13 | 22.73 | 24.25 | 65.00 | **92.34** | 41.69 | 8.42 | 96.31 | **64.30** | 57.28 | 37.47 |
| ReST-KV | 32.01 | 10.73 | 25.23 | 15.91 | 15.85 | 10.25 | 26.47 | 23.23 | 24.79 | 69.00 | 91.62 | 42.59 | 8.40 | 97.66 | 63.48 | 56.03 | **38.33** |
| Full | 32.02 | 13.12 | 27.52 | 16.60 | 16.41 | 11.41 | 34.59 | 23.41 | 26.89 | 73.00 | 91.65 | 43.80 | 7.18 | 97.73 | 65.12 | 58.89 | 39.96 |
| *Mistral-7B-Instruct-v0.3, $B_{total} = 64L$* | | | | | | | | | | | | | | | | | |
| StreamingLLM | 20.37 | 20.56 | 24.62 | 38.87 | 32.47 | 17.68 | 15.48 | 19.84 | 15.81 | 39.50 | 82.77 | 36.72 | 5.50 | 80.00 | 49.77 | 47.90 | 34.24 |
| H2O | 20.51 | 21.52 | 25.12 | 40.12 | 33.12 | 18.34 | 16.23 | 19.12 | 16.24 | 38.50 | 83.12 | 37.23 | 6.00 | 85.50 | 50.12 | 48.12 | 34.93 |
| TOVA | 22.51 | 22.24 | 37.23 | 41.12 | 34.10 | 19.52 | 17.21 | 19.23 | 16.27 | 38.50 | 85.12 | 38.51 | 6.50 | 86.50 | 51.04 | 48.42 | 36.50 |
| SnapKV | 19.39 | 23.62 | 38.66 | 43.26 | 34.72 | 21.33 | 17.59 | 20.93 | 17.06 | 38.50 | 86.96 | 39.61 | **7.00** | 90.50 | 51.63 | 49.73 | 37.53 |
| ReST-KV | **25.65** | **26.58** | **42.71** | **46.11** | **36.43** | **24.34** | **19.80** | **21.65** | **18.90** | **51.50** | **87.88** | **41.54** | 4.00 | 90.50 | **52.39** | **50.75** | **40.05** |
| *Mistral-7B-Instruct-v0.3, $B_{total} = 512L$* | | | | | | | | | | | | | | | | | |
| StreamingLLM | 24.19 | 25.97 | 30.14 | 40.75 | 31.90 | 17.35 | 22.18 | 20.30 | 23.22 | 65.50 | 86.95 | 43.75 | **6.00** | 81.00 | 59.35 | 56.36 | 39.68 |
| H2O | 25.23 | 30.41 | 40.32 | 42.52 | 35.23 | 18.23 | 24.23 | 21.24 | 23.21 | 66.50 | 86.71 | 43.15 | 5.00 | 82.52 | 60.13 | 58.15 | 41.42 |
| TOVA | 25.23 | 32.52 | 46.24 | 45.23 | 36.23 | 20.32 | 24.53 | 22.53 | 23.64 | 66.50 | 87.24 | 44.21 | 6.00 | 85.62 | 59.35 | 60.24 | 42.85 |
| SnapKV | 26.84 | 35.51 | 53.12 | **49.56** | 37.72 | 26.54 | 25.06 | 24.03 | 24.76 | 67.50 | 89.36 | 44.82 | 5.50 | 98.50 | **60.44** | **61.22** | 45.66 |
| ReST-KV | **28.60** | **35.86** | **53.37** | 49.13 | **38.70** | 27.94 | 26.05 | 24.37 | 25.09 | 73.50 | 89.66 | **46.27** | 5.50 | 98.50 | 60.13 | 60.84 | **46.47** |
| Full | 29.07 | 41.54 | 52.88 | 49.37 | 39.01 | 28.58 | 35.07 | 25.71 | 27.73 | 76.00 | 88.59 | 47.51 | 6.00 | 98.50 | 61.48 | 62.68 | 48.11 |
| *Llama2-7B-Chat, $B_{total} = 64L$* | | | | | | | | | | | | | | | | | |
| StreamingLLM | 5.61 | 15.51 | 6.42 | 14.14 | 16.77 | 1.36 | 12.09 | 16.46 | 12.83 | 17.25 | 15.12 | 10.93 | 4.50 | 3.00 | 22.00 | 15.24 | 11.83 |
| H2O | 4.46 | 12.14 | 8.85 | 12.11 | 13.34 | 2.36 | 13.06 | 16.63 | 16.89 | 19.50 | 20.69 | 10.45 | 2.70 | 3.00 | 26.50 | 16.06 | 12.42 |
| TOVA | 8.26 | 14.34 | 12.64 | 13.52 | 13.25 | 3.53 | 11.64 | 16.67 | 13.35 | 36.00 | 72.64 | 32.72 | 2.00 | 4.00 | 36.15 | 32.53 | 20.20 |
| SnapKV | 10.83 | 16.38 | 17.53 | 22.81 | 23.24 | 5.06 | 13.12 | 18.38 | 14.17 | 34.50 | 69.45 | 33.43 | 5.50 | **7.00** | 39.99 | 36.04 | 22.96 |
| LaCache | 8.61 | 16.51 | 7.42 | 15.14 | 17.77 | 4.36 | 13.09 | 17.46 | 13.83 | 18.25 | 18.12 | 12.93 | 5.50 | 6.00 | 24.00 | 17.24 | 13.51 |
| ReST-KV | **12.72** | **17.17** | **24.09** | **24.71** | **23.80** | 5.55 | **15.18** | **19.71** | 17.45 | **43.50** | **76.17** | 33.42 | 5.50 | 4.00 | **45.00** | **40.61** | **25.54** |
| *Llama2-7B-Chat, $B_{total} = 512L$* | | | | | | | | | | | | | | | | | |
| StreamingLLM | 15.30 | 15.53 | 20.16 | 26.59 | 25.05 | 5.65 | 18.30 | 19.28 | 21.84 | 54.50 | 82.23 | 38.07 | 5.50 | 5.00 | 56.80 | 51.95 | 28.86 |
| H2O | 9.68 | 8.67 | 6.86 | 10.85 | 8.71 | 1.31 | 20.04 | 18.72 | **24.91** | 18.00 | 17.09 | 18.99 | 3.75 | 2.30 | 20.87 | 14.87 | 12.85 |
| TOVA | 13.53 | 15.46 | 26.44 | 26.12 | **31.02** | 7.12 | 18.25 | 18.64 | 22.34 | 62.50 | 83.10 | **40.61** | 3.00 | 8.00 | 56.14 | 51.53 | 30.24 |
| SnapKV | 16.22 | 19.57 | 32.32 | 31.87 | 24.97 | 9.66 | 20.19 | 20.77 | 23.85 | 62.00 | 82.24 | 39.18 | **6.00** | 10.50 | 59.49 | 56.06 | 32.18 |
| ReST-KV | **17.15** | **19.88** | **32.71** | **31.94** | 25.62 | 9.97 | **20.52** | 20.68 | 23.59 | **63.50** | **83.30** | 39.29 | 6.00 | **11.50** | 58.65 | 53.81 | **32.38** |

**Implementation Details.** We evaluate ReST-KV and all baselines under varying cache budgets ($B_{total} = nL$, with $n \in [64, 1024]$), where $n$ denotes the number of KV pairs per layer across $L$ layers. To ensure fairness, token eviction is performed only once during the prefilling phase. All methods, except TOVA, are implemented based on the codebase from (Cai, 2023). Experiments are run on NVIDIA A800 80GB GPUs. Further details are provided in Appendix B.

## 4.2 EVALUATIONS ON LONGBENCH DATASET

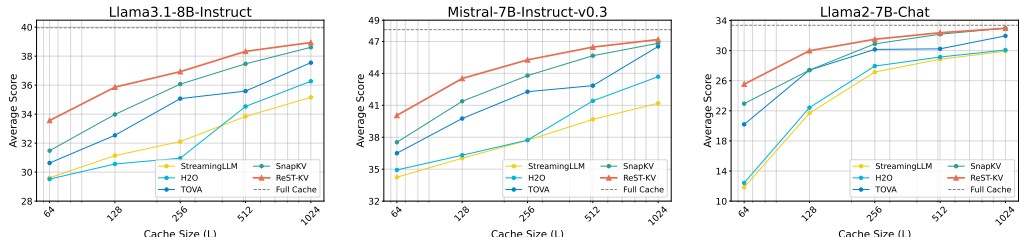

Figure 4: Average score across 16 datasets of LongBench under various cache budgets. ReST-KV outperforms the baseline across different models and settings.

We evaluate ReST-KV on 16 datasets from LongBench. As shown in Figure 4, ReST-KV consistently outperforms all baselines across different cache budget settings, with especially strong gains under tight memory constraints. Unlike prior methods that rely solely on the rank of query-key similarities, our approach accounts the impact of attention redistribution, ensuring that the most critical information is retained. Moreover, we verify the compatibility of ReST-KV with non-uniform bud-

get strategies such as PyramidKV and AdaKV, with results presented in Appendix C. Compatibility with KV cache quantization techniques is also evaluated, as shown in Appendix I.

Table 1 provides a detailed comparison under two cache budgets: low ($B_{\text{total}} = 64L$) and high ($B_{\text{total}} = 512L$), with full results in Appendix D.1. ReST-KV consistently ranks among the top performers across tasks, achieving up to a 2.58% improvement under low budgets with the Mistral model. Notably, ReST-KV substantially outperforms the recent LaCache (Shi et al., 2025) baseline, with a particularly large gap at the tightest budget ($B_{\text{total}} = 64L$, Llama2-7B: 25.54 vs. 13.51). These results highlight the effectiveness of our eviction indicator and spatio-temporal smoothing in enhancing KV selection robustness. Additional evaluations across different models and sizes further confirm this conclusion (Appendix D.2, D.3, D.4).

## 4.3 EVALUATIONS ON RULER BENCHMARK

We evaluate ReST-KV on 11 tasks from the RULER benchmark using the Llama3.1-8B-Instruct model, with a fixed cache budget of $B_{\text{total}} = 1024L$ applied across all methods. Table 2 summarizes the average accuracy across varying context lengths, from 4k to 128k context length. Existing KV cache eviction methods suffer from substantial performance degradation as the context length increases, highlighting their limited robustness in long-context and complex retrieval scenarios. In contrast, ReST-KV consistently achieves strong results across all lengths, with an average accuracy improvement of 15.2% over prior methods. Notably, even at the 128k context lengthwhere less than 1% of the original cache is retained, ReST-KV maintains effective retrieval capabilities. Detailed results for individual tasks are provided in Appendix E.

Table 2: Performance comparison on RULER benchmark across different context lengths.

| Method | 4K | 8K | 16K | 32K | 64K | 128K | Avg. |
|---|---|---|---|---|---|---|---|
| Full | 99.34 | 98.83 | 98.55 | 94.89 | 89.85 | 79.32 | 93.46 |
| Streaming | 39.81 | 18.42 | 12.10 | 10.57 | 9.91 | 8.18 | 16.50 |
| SnapKV | 83.60 | 75.54 | 71.12 | 66.95 | 57.47 | 47.99 | 67.11 |
| PyramidKV | 81.35 | 73.66 | 70.23 | 69.83 | 57.84 | 48.93 | 66.97 |
| **ReST-KV** | **94.01** | **86.66** | **84.12** | **81.87** | **78.65** | **68.28** | **82.27** |

Table 3: Ablation results of ReST-KV.

| Method | Avg. Acc |
|---|---|
| Attention weight Top-k | 32.98 |
| **ReST-KV** | **35.86** |
| ReST-KV w/o LOR | 33.95 (-1.91) |
| ReST-KV w/o EMA | 34.02 (-1.84) |
| ReST-KV w/o AWS | 33.50 (-2.36) |

## 4.4 VISUALIZATION ON NEEDLE-IN-A-HAYSTACK TEST

The needle-in-a-haystack test (Liu et al., 2024a) involves inserting key information at random positions within long contexts and serves as a benchmark to assess the ability of LLMs to accurately retrieve critical information. To further demonstrate the effectiveness and adaptability of our method, we conducted experiments on the Mistral-7B-Instruct-v0.3 model with a cache budget set to $B_{\text{total}} = 1024L$. As shown in Figure 5, even under such a strict cache budget, ReST-KV maintains 98% of the model's performance, significantly outperforming other methods. This underscores ReST-KV's ability to efficiently prioritize and retain the most relevant KV pairs. Additional visualization graphs can be found in Appendix F.

## 4.5 ABLATION STUDIES

We conduct ablation studies on LongBench to evaluate the contribution of each component in our KV cache management strategy: layer-wise output reconstruction (LOR) indicator, exponential moving average (EMA) temporal smoothing, and adaptive window-based spatial smoothing (AWS). We adopt the Llama3.1-8B-Instruct model with a cache budget of $B_{\text{total}} = 128L$ as the default configuration.

Table 3 systematically presents the results. The baseline using vanilla attention-weight-based top-$k$ selection yields only 32.98 accuracy, as it ignores attention redistribution and fails to capture the spatial-temporal dynamics of KV pairs. In contrast, our ReST-KV framework achieves 35.86 accuracy, representing a significant improvement.

To further understand the effectiveness of each module, we ablate them individually:

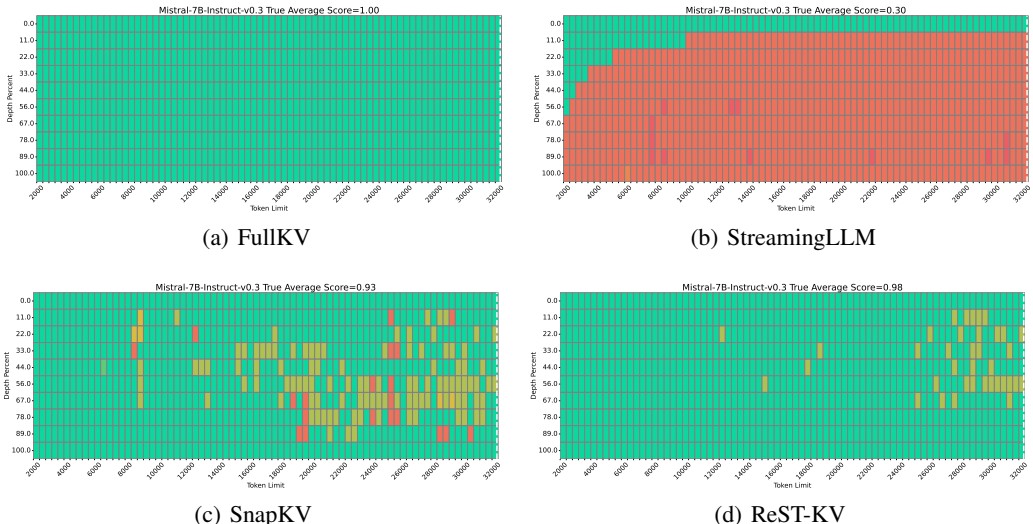

Figure 5: Performance comparison on the Needle in a Haystack Test using Mistral-7B-Instruct-v0.3 with $B_{\text{total}} = 1024L$. Even with a strict cache budget, ReST-KV retains 98% of the model's performance, outperforming other methods in retrieving critical information.

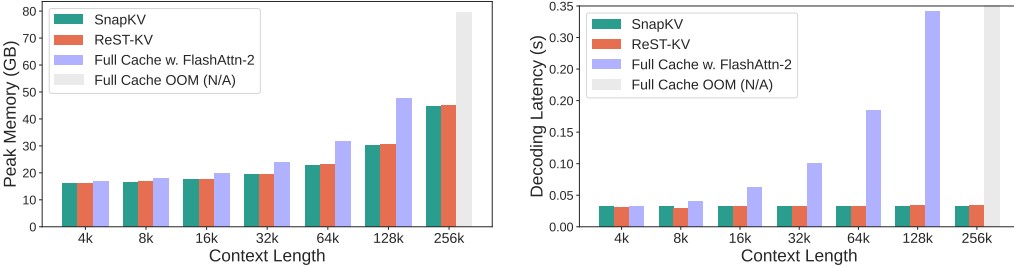

Figure 6: Peak memory usage and decoding latency on NVIDIA A800 80GB GPU. ReST-KV reduces peak memory by 36.0% and achieves up to a 10× speedup at 128k context length compared to full cache.

- Without the LOR indicator, the model misses attention redistribution effects, making it harder to identify truly critical KV pairs. This is especially harmful under tight budgets like $B_{\text{total}} = 128L$, causing a 1.91% drop in accuracy.
- Without EMA temporal smoothing: The model lacks awareness of temporal changes in importance, making it less capable of retaining KV pairs crucial for future queries. This results in a 1.84% performance degradation.
- Without AWS spatial smoothing: Without capturing spatial offset patterns (e.g., vertical-slash structures), the model tends to retain suboptimal KV pairs, causing a 2.36% accuracy drop.

**Hyperparameter Sensitivity.** Figure 7 shows the performance variation with respect to $\alpha$ (left) and $\beta$ (right). Both hyperparameters exhibit stable accuracy across wide ranges, consistently outperforming the baselines. All experiments adopt a fixed setting of $\alpha = 0.3$ and $\beta = 2000$ without any per-model or per-task tuning.

Detailed ablation of each module can be found in Appendix G.

## 4.6 EVALUATION OF MEMORY AND THROUGHPUT

To evaluate the effectiveness and efficiency of our method in reducing memory consumption and enhancing LLM inference, we analyze peak memory usage and decoding latency on the Llama-3.1-8B-Instruct model implemented with FlashAttention-2 (Dao, 2023).

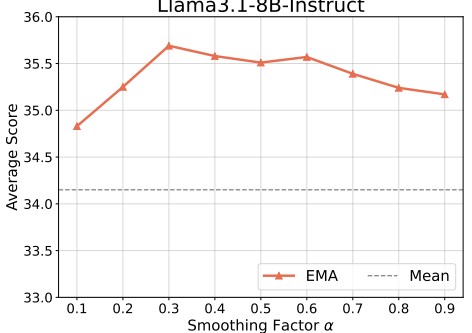 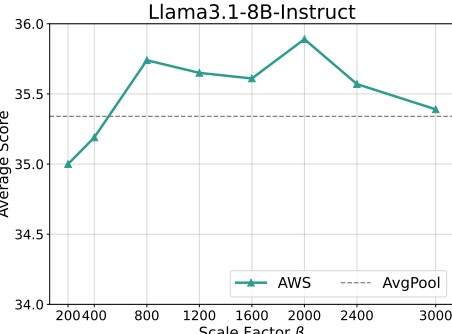

Figure 7: Sensitivity analysis of the smoothing factor $\alpha$ (left) and scaling factor $\beta$ (right). The performance remains relatively stable across different settings of both hyperparameters, mostly outperforming the baseline.

**Peak Memory Usage.** As shown in Figure 6(a), ReST-KV significantly reduces peak memory usage, performing comparably to other KV cache eviction methods. Compared to full cache, ReST-KV achieves approximately 36.0% reduction in peak memory usage at a context length of 128k.

**Latency Analysis.** As shown in Figure 6(b), the decoding latency of the standard full cache method, even with FlashAttention-2, grows rapidly with input length. In contrast, ReST-KV maintains high efficiency by using a fixed cache budget to limit the number of KV pairs. This approach overcomes the latency bottleneck for long sequences, achieving an approximate $10.61\times$ speedup over the full cache method at a 128K context length.

Furthermore, ReST-KV is compatible with prefill sparse attention approaches, yielding a Time-To-First-Token (TTFT) speedup of up to **3.42×**. This efficiency is achieved because our method only requires computing attention outputs within a small query window, resulting in **a computational complexity comparable to that of SnapKV**. For a detailed analysis, please see Appendix H.

**Prefill Overhead.** Although ReST-KV computes an output-aware eviction indicator, it operates only on a small query window of size $w \ll N$, sharing the same $O(wND)$ dominant complexity as SnapKV. Table 4 shows the absolute TTFT under the 128k setting. ReST-KV introduces only $\approx 2\%$ prefill overhead over SnapKVnegligible for long-context workloads where decoding dominates total latency.

Table 4: Prefill TTFT at 128k context length (Llama-3.1-8B-Instruct). ReST-KV incurs only $\approx 2\%$ additional latency over SnapKV, confirming negligible prefill overhead.

| Method | TTFT (ms) |
| --- | --- |
| Full KV | 28,230 |
| SnapKV | 28,510 (+1.0%) |
| **ReST-KV** | **29,082** (+3.0%) |

## 5 CONCLUSION

In this paper, we propose ReST-KV, a novel KV cache eviction method that reformulates eviction as a layer-wise output reconstruction task, effectively capturing attention redistribution effects beyond conventional attention-weight heuristics. To enhance robustness, ReST-KV integrates a spatial-temporal smoothing mechanism using exponential moving averages for temporal stability and adaptive windowing for spatial awareness. Extensive evaluations on LongBench, Needle-in-a-Haystack, and RULER demonstrate that ReST-KV consistently surpasses state-of-the-art methods under low memory budgets and significantly reduces decoding latencyachieving up to $10\times$ speedups at 128k context lengths. Our method is model-agnostic and compatible with existing budget strategies, offering a practical and principled solution for efficient long-context generative inference. Future work will explore tighter integration with adaptive allocation strategies and extensions to multi-modal or structured memory scenarios.

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

## A  DERIVATION AND ANALYSIS OF THE OUTPUT RECONSTRUCTION INDICATOR

We define the eviction indicator $\mathbf{I}_t[n]$ as the reconstruction error of the MHA output caused by removing the $n$-th KV pair. Specifically, the eviction indicator is given by:

$$\mathbf{I}_t[n] = \left\| \mathrm{MHA}\left(\mathbf{x}_t, \langle \mathbf{K}_T, \mathbf{V}_T \rangle\right) - \mathrm{MHA}\left(\mathbf{x}_t, \langle \mathbf{K}_{T,\backslash n}, \mathbf{V}_{T,\backslash n}\rangle\right) \right\|_2, \tag{15}$$

where $\mathbf{K}_{t,\backslash n}$ and $\mathbf{V}_{t,\backslash n}$ represent the set of cache keys and values with the $n$-th KV pair removed.

Using Eq. 3 and Eq. 4, we can expand Eq. 15 as follows:

$$\mathbf{I}_t[n] = \left\| \mathbf{A}_t \mathbf{V}_T \mathbf{W}_O - \mathbf{A}_{t,\backslash n} \mathbf{V}_{T,\backslash n} \mathbf{W}_O \right\|_2 \tag{16}$$

where $\mathbf{A}_{t,\backslash n}$ represents the attention weights with the $n$-th KV pair removed, and $\mathbf{V}_{T,\backslash n}$ represents the values corresponding to the remaining cache sets after the removal of the $n$-th KV pair.

Further, we expand the matrix computation into a weighted sum form as:

$$\mathbf{I}_t[n] = \left\| \sum_m \mathbf{A}_t[m] \mathbf{v}_m \mathbf{W}_O - \sum_{m \neq n} \mathbf{A}_{t,\backslash n}[m] \mathbf{v}_m \mathbf{W}_O \right\|_2 \tag{17}$$

where $\mathbf{A}_t[m]$ and $\mathbf{A}_{t,\backslash n}[m]$ represent the attention weights for the $m$-th query in the presence and absence of the $n$-th KV pair, respectively.

Compared to $\mathbf{A}_t[m]$, $\mathbf{A}_{t,\backslash n}[m]$ is missing the component related to $\mathbf{k}_n$ in the denominator. Therefore, the relationship between the two is given by:

$$\mathbf{A}_{t,\backslash n}[m] = \frac{\mathbf{A}_t[m]}{1 - \mathbf{A}_t[n]} \tag{18}$$

Substituting Eq. 18 into Eq. 17 and performing step-by-step simplifications, we get:

$$\mathbf{I}_t[n] = \left\| \sum_m \mathbf{A}_t[m] \mathbf{v}_m \mathbf{W}_O - \sum_{m \neq n} \frac{\mathbf{A}_t[m]}{1 - \mathbf{A}_t[n]} \mathbf{v}_m \mathbf{W}_O \right\|_2, \tag{19}$$

$$= \left\| \sum_m \mathbf{A}_t[m] \mathbf{v}_m \mathbf{W}_O - \left( \sum_m \frac{\mathbf{A}_t[m]}{1 - \mathbf{A}_t[n]} \mathbf{v}_m \mathbf{W}_O - \frac{\mathbf{A}_t[n]}{1 - \mathbf{A}_t[n]} \mathbf{v}_n \mathbf{W}_O \right) \right\|_2, \tag{20}$$

$$= \left\| \underbrace{\frac{\mathbf{A}_t[n]}{1 - \mathbf{A}_t[n]} \mathbf{v}_n \mathbf{W}_O}_{\text{the } n\text{-th } KV \text{ pair removed's loss}} - \underbrace{\sum_m \frac{\mathbf{A}_t[n]}{1 - \mathbf{A}_t[n]} \cdot \mathbf{A}_t[m] \mathbf{v}_m \mathbf{W}_O}_{\text{the increase of other components after removing the } n\text{-th } KV \text{ pair}} \right\|_2, \tag{21}$$

$$= \frac{\mathbf{A}_t[n]}{1 - \mathbf{A}_t[n]} \cdot \left\| \mathbf{v}_n \mathbf{W}_O - \sum_m \mathbf{A}_t[m] \mathbf{v}_m \mathbf{W}_O \right\|_2, \tag{22}$$

$$= \frac{\mathbf{A}_t[n]}{1 - \mathbf{A}_t[n]} \cdot \left\| \mathrm{MHA}\left(\mathbf{x}_t, \langle \mathbf{K}_T, \mathbf{V}_T \rangle\right) - \mathbf{v}_n \mathbf{W}_O \right\|_2, \tag{23}$$

From Eq. 21, we can see that the layer-wise output reconstruction indicator can be divided into two parts. One part is the loss due to the removal of the $n$-th KV pair, and the other part is the increase in the contribution of the other components after removing the $n$-th KV pair. Together, these two parts determine the importance of a KV pair.

## B  MORE IMPLEMENTATION DETAILS

In this section, we provide additional details regarding the implementation of ReST-KV. Our method operates in two main phases: *prompt prefilling* and *token decoding*. During the prompt prefilling

stage, we employ Eq. 13 from Section 3.3 as the eviction indicator. This formula integrates both the layer-wise output reconstruction indicator and spatial-temporal smoothing. According to Eq. 14, we select a set of KV pairs based on the cache budget from the prompt. Specifically, for the Exponential Moving Average (EMA) Temporal Smoothing, the smoothing factor $\alpha$ is set to 0.3. In the case of the Adaptive Window-Based Spatial Smoothing, the scaling factor $\beta$ is set to 2000. Following SnapKV (Li et al., 2024b), we adopt a fixed observation window of size $S_w = 32$ and kernel size $k = 5$ for SnapKV, PyramidKV, and our proposed ReST-KV. To better capture important information, we set the kernel size to 21 on the RULER and InfiniteBench datasets. The StreamingLLM method retains the first 4 tokens as an attention sink, ensuring efficient processing within the token flow. In the token decoding phase, we utilize the KV cache compressed during the prefilling stage, along with a newly updated KV cache, to perform decoding. Notably, no further compression is applied during this phase. This prefill-only eviction strategy is adopted for consistency with existing methods such as SnapKV and PyramidKV, enabling a fair and controlled comparison. ReST-KV can be straightforwardly extended to decoding-time eviction by periodically re-evaluating KV importance every $M$ generated tokens, which is feasible since the required attention statistics are already available from the decoding pass.

## C  COMPATIBILITY WITH BUDGET ALLOCATION STRATEGIES

In this section, we evaluate the compatibility of our method with existing budget allocation strategies. Specifically, we choose PyramidKV (Cai et al., 2024) as a representative of layer-wise budget allocation strategies and AdaKV (Feng et al., 2024) as a representative of head-wise budget allocation strategies. We compared the average accuracy results of the Llama2-7B-Chat model on the LongBench datasets under varying total cache budgets (ranging from $64L$ to $1024L$). Our experiments demonstrate that, when combined with these strategies, our method achieves similar or slightly improved performance compared to SnapKV combined with the same strategies.

Table 5: Performance comparison of SnapKV and our method with Pyramid layer-wise budget allocation strategies across varying cache budgets.

| Method | Cache Budget $B_{\text{total}}$ | | | | | Avg. Acc |
|---|---|---|---|---|---|---|
| | $64L$ | $128L$ | $256L$ | $512L$ | $1024L$ | |
| SnapKV | 22.96 | 28.31 | 30.90 | 32.18 | 32.99 | 29.47 |
| PyramidKV | 24.67 | 29.58 | 31.04 | 32.32 | 32.95 | 30.11 (↑ 0.64%) |
| ReST-KV | 25.54 | 29.99 | 31.51 | 32.38 | 32.97 | 30.48 |
| ReST-KV w. Pyramid | **26.88** | **30.47** | **31.74** | **32.48** | **33.05** | **30.93** (↑ 0.45%) |

Table 5 illustrates the results of applying Pyramid layer-wise budget allocation strategies to both SnapKV and our method, comparing the performance differences before and after the addition of the strategy. As shown, the accuracy improvements are modest but consistent across different cache budget sizes. For instance, our method combined with layer-wise budget allocation strategies achieves a 0.45% increase in average accuracy across different cache budgets.

Table 6: Performance comparison of SnapKV and our method with Ada head-wise budget allocation strategies across varying cache budgets.

| Method | Cache Budget $B_{\text{total}}$ | | | | | Avg. Acc |
|---|---|---|---|---|---|---|
| | $64L$ | $128L$ | $256L$ | $512L$ | $1024L$ | |
| SnapKV | 22.96 | 28.31 | 30.90 | 32.18 | 32.99 | 29.47 |
| Ada-SnapKV | 24.89 | 29.93 | 31.21 | 32.28 | 33.01 | 30.26 (↑ 0.79%) |
| ReST-KV | 25.54 | 29.99 | 31.51 | 32.38 | 32.97 | 30.48 |
| Ada-ReST-KV | **27.35** | **31.27** | **31.84** | **32.51** | **33.02** | **31.20** (↑ 0.72%) |

Table 6 presents the results of applying head-wise budget allocation strategies to both SnapKV and our method, comparing the performance differences before and after the addition of the strategy. The results show that our method combined with AdaKV achieves a 0.72% increase in average accuracy

Table 7: Performance comparison across 16 datasets of LongBench on Llama3.1-8B-Instruct for cache budgets from $64L$ to $1024L$. The best result is highlighted in **bold**, and the second-best is underlined.

| Method | Single-Document QA | | | Multi-Document QA | | | Summarization | | | Few-shot Learning | | | Synthetic | | Code | | |
| | NrtvQA | Qasper | MF-en | HotpotQA | 2WikiMQA | Musique | GovReport | QMSum | MultiNews | TREC | TriviaQA | SAMSum | PCount | PRe | Lcc | RB-P | Avg. |
|---|---|---|---|---|---|---|---|---|---|---|---|---|---|---|---|---|---|
| *Llama3.1-8B-Instruct, $B_{total} = Full$* | | | | | | | | | | | | | | | | | |
| Full | 32.02 | 13.12 | 27.52 | 16.60 | 16.41 | 11.41 | 34.59 | 23.41 | 26.89 | 73.00 | 91.65 | 43.80 | 7.18 | 97.73 | 65.12 | 58.89 | 39.96 |
| *Llama3.1-8B-Instruct, $B_{total} = 64L$* | | | | | | | | | | | | | | | | | |
| StreamingLLM | 7.65 | 5.08 | 14.14 | 10.93 | 12.64 | 6.86 | 16.57 | 18.93 | 16.30 | 38.50 | 83.13 | 34.65 | **9.78** | **96.28** | 54.16 | 48.21 | 29.61 |
| H2O | 12.23 | 5.12 | 15.12 | 11.51 | 10.14 | 6.23 | 17.23 | 19.51 | 16.79 | 39.15 | 81.51 | 36.12 | 8.12 | 95.12 | 51.25 | 47.12 | 29.52 |
| TOVA | 18.52 | 6.12 | 17.32 | 12.15 | 12.51 | 7.35 | 16.24 | 20.41 | 16.34 | 38.41 | 82.61 | 36.16 | 8.14 | 95.23 | 55.21 | 47.35 | 30.63 |
| SnapKV | 19.90 | 5.78 | 18.38 | 13.51 | 14.42 | **8.52** | 17.35 | 20.44 | 17.33 | 41.00 | 85.37 | 37.63 | 8.93 | 91.08 | 55.09 | **48.88** | 31.48 |
| ReST-KV | **22.43** | **7.19** | **19.25** | **14.11** | **15.04** | 7.97 | **20.56** | **21.10** | **19.15** | **53.50** | **88.23** | **40.21** | 8.46 | 93.90 | **56.74** | 48.77 | **33.54** |
| *Llama3.1-8B-Instruct, $B_{total} = 128L$* | | | | | | | | | | | | | | | | | |
| StreamingLLM | 16.07 | 5.34 | 14.82 | 11.01 | 12.38 | 6.61 | 17.99 | 19.06 | 18.69 | 40.50 | 85.57 | 38.24 | 9.20 | 94.11 | 58.97 | 49.70 | 31.14 |
| H2O | 14.00 | 5.45 | 16.62 | 12.83 | 10.87 | 6.94 | 17.29 | 20.88 | 16.96 | 40.27 | 82.15 | 37.61 | 9.12 | 96.13 | 52.13 | 48.16 | 30.46 |
| TOVA | 21.63 | 8.11 | 18.70 | 14.31 | 14.44 | 9.46 | 19.22 | **22.97** | 17.60 | 40.76 | 84.40 | 39.21 | **11.24** | **96.67** | 58.25 | 48.91 | 32.87 |
| SnapKV | 25.20 | 7.23 | 20.89 | 13.60 | 14.61 | 8.49 | 20.95 | 21.42 | 21.28 | 48.00 | 89.38 | 40.08 | 7.29 | 93.78 | **59.31** | **52.12** | 33.98 |
| ReST-KV | **27.88** | **8.29** | **22.22** | **14.65** | **14.70** | 9.32 | **22.26** | 22.95 | **22.16** | **65.00** | **91.03** | **41.26** | 8.20 | 93.59 | 58.78 | 51.50 | **35.86** |
| *Llama3.1-8B-Instruct, $B_{total} = 256L$* | | | | | | | | | | | | | | | | | |
| StreamingLLM | 16.03 | 5.50 | 14.96 | 10.38 | 12.25 | 7.01 | 20.38 | 19.48 | 20.63 | 46.00 | 87.49 | 41.02 | 9.57 | 90.53 | 61.13 | 51.44 | 32.11 |
| H2O | 13.99 | 6.48 | 17.76 | 13.41 | 11.10 | 7.38 | 17.64 | 21.74 | 18.21 | 40.29 | 82.22 | 38.11 | 8.90 | 96.89 | 51.53 | 49.14 | 30.92 |
| TOVA | 24.05 | **11.17** | 21.30 | **17.61** | **17.50** | **12.84** | 21.93 | **26.16** | 20.58 | 43.69 | 87.29 | **42.52** | **14.21** | **99.26** | 60.92 | 51.65 | 35.79 |
| SnapKV | 27.83 | 9.12 | 22.21 | 13.68 | 14.52 | 10.20 | 23.02 | 23.14 | 22.51 | 56.50 | 90.63 | 40.79 | 7.89 | 97.56 | **62.05** | **55.47** | 36.07 |
| ReST-KV | **29.14** | 9.54 | **23.61** | 14.27 | 14.61 | 9.31 | **24.32** | 23.59 | **23.47** | **67.00** | **92.13** | 42.04 | 8.09 | 94.51 | 61.56 | 53.62 | **36.93** |
| *Llama3.1-8B-Instruct, $B_{total} = 512L$* | | | | | | | | | | | | | | | | | |
| StreamingLLM | 19.15 | 6.47 | 15.02 | 10.94 | 12.58 | 6.23 | 23.66 | 20.05 | 23.31 | 57.50 | 87.70 | 41.86 | **10.25** | 90.74 | 62.39 | 53.61 | 33.84 |
| H2O | 26.23 | 7.34 | 20.51 | 11.52 | 13.52 | 7.34 | 23.23 | 21.24 | 23.16 | 58.50 | 86.12 | 40.15 | 7.25 | 91.02 | 61.23 | 54.12 | 34.53 |
| TOVA | 27.34 | 8.34 | 22.45 | 12.25 | 14.51 | 8.42 | 24.23 | 22.13 | 22.25 | 58.50 | 89.31 | 40.51 | 8.24 | 93.14 | 62.23 | 55.61 | 35.59 |
| SnapKV | 28.02 | 9.83 | 24.84 | 13.77 | 15.40 | 10.21 | 25.13 | 22.73 | 24.25 | 65.00 | **92.34** | 41.69 | 8.42 | 96.31 | **64.30** | **57.28** | 37.47 |
| ReST-KV | **32.01** | **10.73** | **25.23** | **15.91** | **15.85** | **10.25** | **26.47** | **23.23** | **24.79** | **69.00** | 91.62 | **42.59** | 8.40 | **97.66** | 63.48 | 56.03 | **38.33** |
| *Llama3.1-8B-Instruct, $B_{total} = 1024L$* | | | | | | | | | | | | | | | | | |
| StreamingLLM | 20.50 | 8.08 | 15.72 | 11.61 | 12.39 | 6.71 | 25.76 | 20.18 | 25.44 | 63.50 | 88.84 | 42.61 | 10.03 | 92.10 | 63.15 | 55.88 | 35.16 |
| H2O | 27.63 | 8.84 | 21.98 | 12.99 | 15.91 | 8.23 | 23.96 | 23.77 | 24.20 | 59.79 | 86.97 | 41.52 | 9.07 | 93.01 | 63.59 | 56.08 | 36.10 |
| TOVA | 29.82 | 9.73 | 25.10 | 14.92 | 17.53 | 10.20 | 27.06 | 23.20 | 24.78 | 59.89 | **92.21** | **43.49** | **10.38** | 95.86 | 64.08 | 57.47 | 37.86 |
| SnapKV | **31.95** | 11.26 | 25.56 | 15.13 | 16.18 | 10.79 | 26.97 | 23.06 | 25.89 | 67.50 | 91.90 | 42.88 | 7.67 | **98.16** | **64.53** | **58.30** | 38.61 |
| ReST-KV | 31.83 | **11.61** | **26.51** | **15.85** | 15.48 | **10.83** | **28.20** | **24.00** | **26.18** | **70.50** | 91.73 | 42.70 | 8.02 | 97.79 | 64.24 | 57.56 | **38.94** |

across all cache budgets. These results highlight that our method is compatible with existing budget allocation strategies.

# D    ADDITIONAL EXPERIMENTS ON LONGBENCH

In this section, we provide comprehensive experimental results on LongBench (Bai et al., 2023), a benchmark focused on long-context understanding, with input lengths ranging from 1235 to 18409 tokens. We perform detailed performance evaluations for three base models with cache budgets ranging from $64L$ to $1024L$: Llama2-7B-Chat (Touvron et al., 2023), Llama3.1-8B-Instruct (Dubey et al., 2024), and Mistral-7B-Instruct-v0.3 (Jiang et al., 2023) (Appendix D.1). To demonstrate the generality of ReST-KV, we also conduct experiments across different models and sizes. In Appendix D.2, we report additional experiments on the Qwen2.5-7B-Instruct (Team, 2024) and Gemma-7B-Instruct (Team et al., 2024) model architectures, and in Appendix D.3, we present experiments on the Llama2-13B-Chat and Llama3-70B-Instruct model sizes.

## D.1    DETAILED PERFORMANCE ACROSS CACHE BUDGETS

Tables 7, 8, and 9 present the detailed LongBench results of ReST-KV and comparative methods applied to Llama3.1-8B-Instruct, Mistral-7B-Instruct-v0.3, and Llama2-7B-Chat, respectively. Overall, the results demonstrate that, compared to other methods, ReST-KV consistently outperforms all baselines across all tasks in LongBench when applied to the test models with cache budgets ranging from $64L$ to $1024L$. This proves the effectiveness and wide applicability of ReST-KV in efficient long-context processing using KV caches in open-source LLMs across domains.

Table 8: Performance comparison across 16 datasets of LongBench on Mistral-7B-Instruct-v0.3 for cache budgets from $64L$ to $1024L$. The best result is highlighted in **bold**, and the second-best is underlined.

| Method | Single-Document QA | | | Multi-Document QA | | | Summarization | | | Few-shot Learning | | | Synthetic | | Code | | |
| | NrtvQA | Qasper | MF-en | HotpotQA | 2WikiMQA | Musique | GovReport | QMSum | MultiNews | TREC | TriviaQA | SAMSum | PCount | PRe | Lcc | RB-P | Avg. |
|---|---|---|---|---|---|---|---|---|---|---|---|---|---|---|---|---|---|
| | Mistral-7B-Instruct-v0.3, $B_{\text{total}} = Full$ | | | | | | | | | | | | | | | | |
| Full | 29.07 | 41.54 | 52.88 | 49.37 | 39.01 | 28.58 | 35.07 | 25.71 | 27.73 | 76.00 | 88.59 | 47.51 | 6.00 | 98.50 | 61.48 | 62.68 | 48.11 |
| | Mistral-7B-Instruct-v0.3, $B_{\text{total}} = 64L$ | | | | | | | | | | | | | | | | |
| StreamingLLM | 20.37 | 20.56 | 24.62 | 38.87 | 32.47 | 17.68 | 15.48 | 19.84 | 15.81 | 39.50 | 82.77 | 36.72 | 5.50 | 80.00 | 49.77 | 47.90 | 34.24 |
| H2O | 20.51 | 21.52 | 25.12 | 40.12 | 33.12 | 18.34 | 16.23 | 19.12 | 16.24 | 38.50 | 83.12 | 37.23 | 6.00 | 85.50 | 50.12 | 48.12 | 34.93 |
| TOVA | 22.51 | 22.24 | 37.23 | 41.12 | 34.10 | 19.52 | 17.21 | 19.23 | 16.27 | 38.50 | 85.12 | 38.51 | 6.50 | 86.50 | 51.04 | 48.42 | 36.50 |
| SnapKV | 19.39 | 23.62 | 38.66 | 43.26 | 34.72 | 21.33 | 17.59 | 20.93 | 17.06 | 38.50 | 86.96 | 39.61 | 7.00 | 90.50 | 51.63 | 49.73 | 37.53 |
| ReST-KV | 25.65 | 26.58 | 42.71 | 46.11 | 36.43 | 24.34 | 19.80 | 21.65 | 18.90 | 51.50 | 87.88 | 41.54 | 4.00 | 90.50 | 52.39 | 50.75 | 40.05 |
| | Mistral-7B-Instruct-v0.3, $B_{\text{total}} = 128L$ | | | | | | | | | | | | | | | | |
| StreamingLLM | 21.39 | 22.05 | 26.73 | 37.25 | 32.81 | 17.61 | 16.76 | 19.69 | 17.98 | 45.50 | 85.64 | 40.49 | 5.50 | 80.00 | 55.01 | 52.12 | 36.03 |
| H2O | 22.39 | 22.98 | 26.92 | 41.12 | 33.19 | 19.20 | 16.90 | 20.70 | 16.82 | 41.12 | 87.10 | 39.76 | 8.58 | 86.19 | 50.81 | 53.01 | 36.76 |
| TOVA | 22.48 | 28.78 | 48.71 | 47.58 | 34.26 | 21.96 | 21.67 | 21.75 | 21.68 | 42.23 | 87.04 | 42.10 | 2.08 | 94.58 | 56.97 | 54.76 | 40.54 |
| SnapKV | 25.04 | 28.42 | 47.88 | 46.23 | 36.47 | 24.60 | 21.22 | 22.74 | 21.15 | 45.00 | 88.74 | 43.07 | 4.00 | 95.00 | 56.81 | 55.75 | 41.38 |
| ReST-KV | 26.58 | 29.60 | 49.23 | 47.46 | 37.18 | 25.16 | 22.44 | 22.43 | 21.77 | 69.00 | 88.18 | 43.84 | 5.50 | 96.50 | 56.29 | 55.13 | 43.52 |
| | Mistral-7B-Instruct-v0.3, $B_{\text{total}} = 256L$ | | | | | | | | | | | | | | | | |
| StreamingLLM | 22.46 | 23.32 | 29.63 | 39.62 | 32.01 | 16.71 | 19.13 | 19.30 | 20.14 | 54.50 | 85.12 | 43.21 | 5.50 | 80.00 | 57.72 | 55.03 | 37.71 |
| H2O | 24.31 | 23.78 | 27.97 | 43.90 | 33.95 | 19.87 | 17.42 | 23.36 | 17.32 | 43.74 | 91.10 | 40.17 | 11.88 | 86.92 | 51.54 | 54.24 | 38.22 |
| TOVA | 28.17 | 29.93 | 51.01 | 46.26 | 36.55 | 26.65 | 22.76 | 22.31 | 21.24 | 54.34 | 88.00 | 42.45 | 2.16 | 94.49 | 57.39 | 58.04 | 42.61 |
| SnapKV | 26.88 | 31.72 | 51.40 | 48.89 | 36.80 | 27.33 | 22.85 | 23.66 | 23.15 | 57.00 | 89.01 | 43.60 | 5.00 | 96.50 | 58.64 | 58.21 | 43.79 |
| ReST-KV | 27.43 | 34.24 | 52.11 | 48.81 | 38.25 | 27.20 | 24.31 | 23.33 | 23.24 | 72.50 | 88.59 | 44.61 | 5.50 | 96.50 | 58.41 | 59.21 | 45.27 |
| | Mistral-7B-Instruct-v0.3, $B_{\text{total}} = 512L$ | | | | | | | | | | | | | | | | |
| StreamingLLM | 24.19 | 25.97 | 30.14 | 40.75 | 31.90 | 17.35 | 22.18 | 20.30 | 23.22 | 65.50 | 86.95 | 43.75 | 6.00 | 81.00 | 59.35 | 56.36 | 39.68 |
| H2O | 25.23 | 30.41 | 40.32 | 42.52 | 35.23 | 18.23 | 24.23 | 21.24 | 23.21 | 66.50 | 86.71 | 43.15 | 5.00 | 82.52 | 60.13 | 58.15 | 41.42 |
| TOVA | 25.23 | 32.52 | 46.24 | 45.23 | 36.23 | 20.32 | 24.53 | 22.53 | 23.64 | 66.50 | 87.24 | 44.21 | 6.00 | 85.62 | 59.35 | 60.24 | 42.85 |
| SnapKV | 26.84 | 35.51 | 53.12 | 49.56 | 37.72 | 26.54 | 25.06 | 24.03 | 24.76 | 67.50 | 89.36 | 44.82 | 5.50 | 98.50 | 60.44 | 61.22 | 45.66 |
| ReST-KV | 28.60 | 35.86 | 53.37 | 49.13 | 38.70 | 27.94 | 26.05 | 24.37 | 25.09 | 73.50 | 89.66 | 46.27 | 5.50 | 98.50 | 60.13 | 60.84 | 46.47 |
| | Mistral-7B-Instruct-v0.3, $B_{\text{total}} = 1024L$ | | | | | | | | | | | | | | | | |
| StreamingLLM | 24.81 | 27.98 | 31.09 | 42.93 | 32.65 | 18.03 | 24.57 | 20.74 | 25.42 | 68.50 | 88.71 | 45.37 | 5.50 | 82.50 | 61.07 | 59.21 | 41.19 |
| H2O | 28.23 | 32.61 | 42.96 | 45.03 | 38.39 | 20.56 | 26.50 | 24.01 | 25.10 | 69.37 | 88.49 | 45.60 | 8.11 | 83.81 | 62.79 | 59.90 | 43.84 |
| TOVA | 29.10 | 36.82 | 53.78 | 49.25 | 38.39 | 28.33 | 27.17 | 23.75 | 25.53 | 70.39 | 88.28 | 45.24 | 4.85 | 100.47 | 60.40 | 62.25 | 46.50 |
| SnapKV | 29.31 | 37.25 | 53.55 | 49.25 | 38.54 | 28.28 | 26.90 | 24.49 | 26.27 | 72.50 | 89.11 | 46.08 | 5.50 | 99.00 | 61.45 | 61.76 | 46.83 |
| ReST-KV | 29.20 | 37.72 | 52.56 | 50.50 | 38.89 | 28.69 | 28.03 | 24.71 | 26.76 | 74.00 | 89.41 | 47.08 | 5.50 | 99.00 | 61.10 | 61.66 | 47.18 |

## D.2 ADDITIONAL EXPERIMENTS ON MORE MODEL ARCHITECTURES

To further validate the versatility of ReST-KV across different model architectures, we performed additional experiments on the Qwen2.5-7B-Instruct and Gemma-7B-Instruct models. The experiments were conducted in two distinct memory configurations: a low-memory setting ($B_{\text{total}} = 64L$) and a high-memory setting ($B_{\text{total}} = 512L$). As shown in Table 10, ReST-KV consistently outperforms baseline methods in both the low and high memory settings for the Qwen and Gemma architectures, similar to the results observed with the Llama and Mistral models. These findings further confirm the adaptability of ReST-KV across various model architectures, demonstrating its robust performance advantage regardless of the underlying design of the models.

## D.3 ADDITIONAL EXPERIMENTS ON LARGER-SCALE MODELS

To assess the scalability of ReST-KV on larger models, we conducted additional experiments on Llama2-13B-Chat and Llama3-70B-Instruct. These experiments were performed under two different memory configurations: a low-memory setting ($B_{\text{total}} = 64L$) and a high-memory setting ($B_{\text{total}} = 512L$). As shown in Table 11, ReST-KV consistently outperforms baseline methods in both low and high memory settings for the Llama2-13B-Chat and Llama3-70B-Instruct models. These results further demonstrate the scalability and effectiveness of ReST-KV when applied to larger-scale models, highlighting its continued performance advantage regardless of the model size.

## D.4 COMPARISON WITH LACACHE

We compare ReST-KV with LaCache (Shi et al., 2025) on Llama2-7B-Chat under percentage-based cache budgets (25% and 50%) across all 16 LongBench tasks. As shown in Tables 12 and 13, ReST-KV consistently outperforms LaCache under both budget settings, with the advantage particularly

Table 9: Performance comparison across 16 datasets of LongBench on Llama2-7B-Chat for cache budgets from $64L$ to $1024$. The best result is highlighted in **bold**, and the second-best is underlined.

| Method | Single-Document QA | | | Multi-Document QA | | | Summarization | | | Few-shot Learning | | | Synthetic | | Code | | Avg. |
|---|---|---|---|---|---|---|---|---|---|---|---|---|---|---|---|---|---|
| | NrtvQA | Qasper | MF-en | HotpotQA | 2WikiMQA | Musique | GovReport | QMSum | MultiNews | TREC | TriviaQA | SAMSum | PCount | PRe | Lcc | RB-P | |
| Llama2-7B-Chat, $B_{\text{total}} = Full$ | | | | | | | | | | | | | | | | | |
| Full | 18.39 | 20.11 | 35.67 | 31.25 | 25.50 | 10.14 | 25.68 | 20.93 | 26.27 | 64.00 | 83.38 | 40.99 | 5.50 | 10.00 | 60.81 | 55.27 | 33.37 |
| Llama2-7B-Chat, $B_{\text{total}} = 64L$ | | | | | | | | | | | | | | | | | |
| StreamingLLM | 5.61 | 15.51 | 6.42 | 14.14 | 16.77 | 1.36 | 12.09 | 16.46 | 12.83 | 17.25 | 15.12 | 10.93 | 4.50 | 3.00 | 22.00 | 15.24 | 11.83 |
| H2O | 4.46 | 12.14 | 8.85 | 12.11 | 13.34 | 2.36 | 13.06 | 16.63 | 16.89 | 19.50 | 20.69 | 10.45 | 2.70 | 3.00 | 26.50 | 16.06 | 12.42 |
| TOVA | 8.26 | 14.34 | 12.64 | 13.52 | 13.25 | 3.53 | 11.64 | 16.67 | 13.35 | 36.00 | 72.64 | 32.72 | 2.00 | 4.00 | 36.15 | 32.53 | 20.20 |
| SnapKV | 10.83 | 16.38 | 17.53 | 22.81 | 23.24 | 5.06 | 13.12 | 18.38 | 14.17 | 34.50 | 69.45 | **33.43** | 5.50 | **7.00** | 39.99 | **36.04** | 22.96 |
| ReST-KV | **12.72** | **17.17** | **24.09** | **24.71** | **23.80** | **5.55** | **15.18** | **19.71** | **17.45** | **43.50** | **76.17** | 33.42 | 5.50 | 4.00 | **45.00** | **40.61** | **25.54** |
| Llama2-7B-Chat, $B_{\text{total}} = 128L$ | | | | | | | | | | | | | | | | | |
| StreamingLLM | 8.45 | 14.87 | 12.68 | 19.98 | 22.14 | 5.17 | 13.99 | 19.74 | 16.02 | 28.50 | 60.96 | 30.61 | 5.00 | 5.00 | 44.44 | 39.53 | 21.69 |
| H2O | 7.60 | 9.53 | 9.92 | 18.35 | 15.64 | 3.30 | 17.75 | 14.71 | **21.45** | 28.00 | 39.61 | 13.85 | 4.17 | 3.56 | 29.92 | 25.53 | 16.43 |
| TOVA | 12.26 | 14.66 | 25.72 | 26.08 | 24.21 | 6.90 | 15.28 | 18.30 | 17.61 | 42.44 | 80.12 | 35.25 | 5.05 | 6.93 | 52.48 | 49.17 | 27.03 |
| SnapKV | 13.32 | 16.28 | 27.23 | 27.23 | 24.37 | 7.17 | 16.97 | 19.65 | 19.38 | 44.00 | 81.88 | 36.82 | 6.00 | 8.00 | **54.02** | **50.66** | 28.31 |
| ReST-KV | **15.55** | **17.78** | **27.24** | **27.72** | **24.62** | **8.93** | **17.88** | **20.13** | 20.92 | **60.00** | **82.48** | **37.35** | 6.00 | **9.50** | 53.45 | 50.24 | **29.99** |
| Llama2-7B-Chat, $B_{\text{total}} = 256L$ | | | | | | | | | | | | | | | | | |
| StreamingLLM | 13.81 | 15.51 | 17.63 | 25.81 | 24.48 | 7.70 | 16.16 | 19.33 | 18.78 | 44.00 | 78.87 | 37.63 | 5.50 | 5.00 | 54.57 | 49.68 | 27.15 |
| H2O | 8.82 | 11.73 | 10.11 | 15.54 | 13.70 | 3.78 | **19.29** | 19.13 | **23.36** | 34.00 | 35.61 | 20.26 | 4.75 | 3.57 | 23.74 | 23.75 | 16.95 |
| TOVA | 14.12 | 16.82 | 29.15 | 27.69 | 24.82 | 6.89 | 18.04 | 18.67 | 21.79 | 57.01 | 83.48 | 37.74 | 5.06 | 8.71 | 56.51 | 52.99 | 29.97 |
| SnapKV | **15.45** | 17.57 | 29.44 | 31.53 | 24.94 | 8.69 | 18.78 | 20.48 | 22.15 | 57.50 | **83.76** | 38.25 | 6.00 | 10.50 | **57.75** | **53.59** | 30.90 |
| ReST-KV | 15.23 | **18.57** | **30.46** | **31.53** | **25.85** | **9.09** | 19.13 | **20.83** | 22.28 | **63.00** | 82.57 | **39.05** | 6.00 | **11.50** | 57.16 | 51.91 | **31.51** |
| Llama2-7B-Chat, $B_{\text{total}} = 512L$ | | | | | | | | | | | | | | | | | |
| StreamingLLM | 15.30 | 15.53 | 20.16 | 26.59 | 25.05 | 5.65 | 18.30 | 19.28 | 21.84 | 54.50 | 82.23 | 38.07 | 5.50 | 5.00 | 56.80 | 51.95 | 28.86 |
| H2O | 9.68 | 8.67 | 6.86 | 10.85 | 8.71 | 1.31 | 20.04 | 18.72 | **24.91** | 18.00 | 17.09 | 18.99 | 3.75 | 2.30 | 20.87 | 14.87 | 12.85 |
| TOVA | 13.53 | 15.46 | 26.44 | 26.12 | **31.02** | 7.12 | 18.25 | 18.64 | 22.34 | 62.50 | 83.10 | **40.61** | 3.00 | 8.00 | 56.14 | 51.53 | 30.24 |
| SnapKV | 16.22 | 19.57 | 32.32 | 31.87 | 24.97 | 9.66 | 20.19 | 20.77 | 23.85 | 62.00 | 82.24 | 39.18 | 6.00 | 10.50 | **59.49** | **56.06** | 32.18 |
| ReST-KV | **17.15** | **19.88** | **32.71** | **31.94** | 25.62 | 9.97 | **20.52** | 20.68 | 23.59 | **63.50** | **83.30** | 39.29 | 6.00 | **11.50** | 58.65 | 53.81 | **32.38** |
| Llama2-7B-Chat, $B_{\text{total}} = 1024L$ | | | | | | | | | | | | | | | | | |
| StreamingLLM | 15.12 | 17.35 | 22.21 | 26.76 | 24.43 | 6.52 | 21.15 | 19.16 | 24.67 | 61.00 | 82.16 | 39.69 | **6.00** | 1.50 | 57.73 | 53.24 | 29.92 |
| H2O | 6.55 | 11.17 | 8.96 | 13.56 | 9.57 | 1.80 | 22.43 | 19.74 | 26.07 | 18.50 | 15.59 | 36.61 | 4.43 | 1.08 | 29.96 | 15.24 | 15.08 |
| TOVA | 16.84 | 19.32 | 34.90 | 31.07 | 25.24 | 9.51 | 20.36 | 20.34 | 23.42 | 62.38 | 81.31 | 39.68 | 4.03 | 10.05 | 58.13 | 54.73 | 31.96 |
| SnapKV | 17.41 | 19.74 | **35.92** | 31.82 | **26.00** | 10.09 | 22.06 | 20.43 | 24.88 | 63.50 | 82.77 | 40.52 | **6.00** | **10.50** | 60.10 | **56.05** | **32.99** |
| ReST-KV | 17.39 | 20.01 | 35.33 | 31.71 | 25.33 | 9.60 | 22.30 | 20.85 | 24.91 | 63.50 | **83.73** | 40.76 | **6.00** | **10.50** | 60.57 | 54.95 | 32.97 |

pronounced at the 25% budget. These results complement the fixed-budget comparison in the main text (Table 1).

# E    ADDITIONAL EXPERIMENTS ON RULER BENCHMARK

In this section, we present a detailed evaluation of ReST-KV on the various subtasks of the RULER benchmark (Hsieh et al., 2024). RULER is specifically designed to assess the core capabilities of LLMs in long-context scenarios through a diverse suite of tasks.

The retrieval suite includes four variants of the needle-in-a-haystack (NIAH) testSingle-Needle (S-NIAH), Multi-Key (MK-NIAH), Multi-Query (MQ-NIAH), and Multi-Value (MV-NIAH)to evaluate recall accuracy under diverse distractor settings and query formulations. Beyond retrieval, the Variable Tracking (VT) task measures multi-hop reasoning by requiring models to resolve transitive variable references scattered throughout the input. Lastly, aggregation tasks such as Common Word Extraction (CWE) and Frequent Word Extraction (FWE) test a model's ability to compress and synthesize high-density signal distributed across long contexts.These tasks collectively pose distinct challenges for context retention, salience estimation, and compositional reasoning, providing a holistic benchmark for evaluating memory management strategies like ReST-KV.

We evaluate ReST-KV using the LLaMA-3.1-8B-Instruct model with a maximum context window of $B = 1024L$, across input lengths ranging from 4k to 128k tokens. The evaluation compares ReST-KV with several representative KV cache eviction baselines: Full KV cache (oracle), StreamingLLM (Xiao et al., 2023), SnapKV (Li et al., 2024b), and PyramidKV (Cai et al., 2024).

As reported in Table 14, ReST-KV consistently achieves higher average accuracy than all alternative eviction strategies across all context lengths. For instance, at 4k tokens, ReST-KV achieves an average accuracy of 94.01%, substantially outperforming SnapKV (83.60%) and PyramidKV (85.21%). While all methods exhibit declining performance as the context length increases, ReST-KV main-

Table 10: Performance comparison across 16 datasets of LongBench on Qwen2.5-7B-Instruct and Gemma-7B-Instruct. The best result is highlighted in **bold**, and the second-best is underlined.

| Method | Single-Document QA | | | Multi-Document QA | | | Summarization | | | Few-shot Learning | | | Synthetic | | Code | | Avg. |
|---|---|---|---|---|---|---|---|---|---|---|---|---|---|---|---|---|---|
| | NrtvQA | Qasper | MF-en | HotpotQA | 2WikiMQA | Musique | GovReport | QMSum | MultiNews | TREC | TriviaQA | SAMSum | PCount | PRe | Lcc | RB-P | |
| Qwen2.5-7B-Instruct, $B_{\text{total}} = Full$ | | | | | | | | | | | | | | | | | |
| Full | 3.82 | 10.75 | 24.24 | 10.23 | 9.30 | 6.97 | 32.54 | 17.84 | 22.46 | 71.50 | 89.32 | 46.16 | 4.35 | 98.83 | 61.93 | 68.2 | 36.15 |
| Qwen2.5-7B-Instruct, $B_{\text{total}} = 64L$ | | | | | | | | | | | | | | | | | |
| StreamingLLM | 2.74 | 5.53 | 13.16 | 7.62 | 7.70 | 4.35 | 14.98 | 12.70 | 11.96 | 38.50 | 77.44 | 37.51 | 6.29 | 26.29 | 44.14 | 44.40 | 22.21 |
| H2O | 1.09 | 3.46 | 17.22 | 6.57 | 6.33 | 4.23 | 14.50 | 11.67 | 10.29 | 37.38 | 76.86 | 37.81 | 7.76 | 83.84 | 46.11 | 49.59 | 25.92 |
| TOVA | 2.19 | 3.58 | 17.34 | 8.23 | 8.14 | 5.96 | 16.48 | 13.38 | 11.36 | 37.94 | 78.75 | 39.12 | 7.79 | 85.47 | 47.93 | 50.24 | 27.12 |
| SnapKV | 2.86 | 5.58 | 18.71 | 8.59 | 8.41 | 6.01 | 16.96 | 13.67 | 13.21 | 39.50 | **79.09** | 40.39 | **7.92** | 87.02 | 48.10 | **51.52** | 27.97 |
| ReST-KV | **3.27** | **6.69** | **18.95** | **9.57** | **8.79** | **6.03** | **18.77** | **14.99** | **15.19** | **50.50** | 79.07 | **41.28** | 4.73 | **93.00** | **48.47** | 50.03 | **29.33** |
| Qwen2.5-7B-Instruct, $B_{\text{total}} = 512L$ | | | | | | | | | | | | | | | | | |
| StreamingLLM | 2.98 | 6.70 | 15.29 | 8.28 | 8.27 | 4.15 | 22.54 | 13.15 | 18.90 | 56.00 | 85.96 | 43.43 | **6.84** | 36.21 | 54.08 | 53.69 | 27.28 |
| H2O | 1.16 | 6.46 | 21.50 | 8.05 | 8.50 | 6.00 | 23.09 | 15.25 | 17.63 | 63.65 | 81.84 | 43.52 | 2.03 | 93.33 | 57.68 | 61.67 | 31.96 |
| TOVA | 2.70 | 7.99 | 21.77 | 8.61 | 8.57 | 6.61 | 23.44 | 16.34 | 19.42 | 64.28 | 82.98 | 44.41 | 2.35 | 94.70 | 59.23 | 63.58 | 32.94 |
| SnapKV | **3.57** | 8.90 | 22.88 | 10.34 | 9.57 | 6.74 | 24.73 | 17.58 | 19.56 | 64.50 | 83.49 | **45.08** | 4.32 | 96.67 | 59.93 | **64.54** | 33.90 |
| ReST-KV | 3.53 | **9.46** | **23.66** | **10.91** | **9.76** | **7.24** | **25.65** | **17.76** | **20.23** | **67.50** | **86.73** | 44.93 | 3.83 | **98.08** | **60.14** | 63.56 | **34.56** |
| Gemma-7B-Instruct, $B_{\text{total}} = Full$ | | | | | | | | | | | | | | | | | |
| Full | 14.28 | 33.12 | 41.08 | 30.75 | 26.11 | 15.47 | 23.95 | 19.31 | 23.86 | 69.50 | 81.28 | 36.22 | 4.00 | 35.92 | 48.47 | 48.79 | 34.51 |
| Gemma-7B-Instruct, $B_{\text{total}} = 64L$ | | | | | | | | | | | | | | | | | |
| StreamingLLM | 11.31 | 16.54 | 22.96 | 21.87 | 23.25 | 10.18 | 12.47 | 16.80 | 12.74 | 38.50 | 70.94 | 29.79 | 2.50 | 20.50 | 44.67 | 48.75 | 25.24 |
| H2O | 10.37 | 15.93 | 33.33 | 26.09 | 23.65 | 12.52 | 12.49 | 16.94 | 12.99 | 39.15 | 80.42 | 32.23 | 3.20 | 24.41 | 46.00 | 49.03 | 27.42 |
| TOVA | 10.53 | 16.58 | 33.81 | 27.05 | 24.56 | 12.66 | 13.26 | 17.19 | 13.76 | 39.68 | 80.69 | 32.67 | 3.68 | 25.29 | 46.01 | 49.94 | 27.96 |
| SnapKV | 11.05 | 17.06 | 34.22 | 27.41 | 25.32 | 13.52 | 13.98 | 17.35 | 14.03 | 40.50 | 81.42 | **32.90** | **3.83** | 26.00 | 46.34 | **49.95** | 28.43 |
| ReST-KV | **13.10** | **22.90** | **36.78** | **28.36** | **25.90** | **15.13** | **15.39** | **18.09** | **15.68** | **44.50** | **82.55** | 32.77 | 3.33 | **36.25** | **47.92** | 48.70 | **30.46** |
| Gemma-7B-Instruct, $B_{\text{total}} = 512L$ | | | | | | | | | | | | | | | | | |
| StreamingLLM | 11.58 | 20.76 | 26.09 | 24.06 | 23.36 | 10.62 | 17.49 | 17.01 | 20.12 | 60.50 | 78.20 | 37.45 | 1.83 | 25.17 | **49.88** | **52.64** | 29.80 |
| H2O | 12.70 | 29.01 | 38.66 | 28.71 | 25.10 | 14.19 | 17.53 | 17.53 | 20.09 | 60.94 | 80.75 | 35.24 | 3.15 | 34.02 | 47.44 | 49.01 | 32.14 |
| TOVA | 13.23 | 29.30 | 39.32 | 29.63 | 25.57 | 14.55 | 18.12 | 18.32 | 21.01 | 61.13 | 81.49 | 35.78 | 3.61 | 34.46 | 48.39 | 50.00 | 32.74 |
| SnapKV | 13.36 | 29.43 | 39.80 | **30.24** | 26.01 | **14.82** | 18.30 | **18.86** | 21.23 | 62.00 | 81.51 | 36.04 | **4.33** | 35.08 | 49.16 | 50.73 | 33.18 |
| ReST-KV | **13.70** | **30.33** | **42.08** | 30.13 | **26.06** | 14.37 | **18.82** | 18.60 | **22.38** | **69.00** | **81.72** | **37.55** | **4.33** | **35.21** | 48.67 | 49.48 | **33.90** |

Table 11: Performance comparison across 16 datasets of LongBench on Llama models from 13B to 70B. The best result is highlighted in **bold**, and the second-best is underlined.

| Method | Single-Document QA | | | Multi-Document QA | | | Summarization | | | Few-shot Learning | | | Synthetic | | Code | | Avg. |
|---|---|---|---|---|---|---|---|---|---|---|---|---|---|---|---|---|---|
| | NrtvQA | Qasper | MF-en | HotpotQA | 2WikiMQA | Musique | GovReport | QMSum | MultiNews | TREC | TriviaQA | SAMSum | PCount | PRe | Lcc | RB-P | |
| Llama2-13B-Chat, $B_{\text{total}} = Full$ | | | | | | | | | | | | | | | | | |
| Full | 19.19 | 25.86 | 37.04 | 36.65 | 33.22 | 14.02 | 25.92 | 20.24 | 26.02 | 65.00 | 87.70 | 35.60 | 3.60 | 11.00 | 51.26 | 53.15 | 34.09 |
| Llama2-13B-Chat, $B_{\text{total}} = 64L$ | | | | | | | | | | | | | | | | | |
| StreamingLLM | 6.95 | 15.50 | 16.08 | 24.30 | 26.66 | 7.49 | 0.98 | 17.89 | 2.17 | 29.50 | 55.63 | 16.70 | 3.00 | 5.50 | 34.05 | 29.27 | 18.23 |
| H2O | 14.00 | 18.17 | 21.78 | 30.62 | 28.77 | 9.93 | 14.88 | 19.17 | 17.97 | 35.00 | 80.11 | 28.97 | **3.87** | 6.50 | 37.50 | 30.11 | 24.83 |
| TOVA | 15.10 | 17.12 | 24.22 | 34.11 | 28.32 | 10.69 | 14.60 | 18.89 | 16.83 | 33.57 | 86.42 | 29.99 | 3.13 | 9.79 | 40.48 | 38.05 | 26.33 |
| SnapKV | **16.05** | 17.20 | 24.85 | 34.51 | 28.72 | **11.52** | 15.39 | **19.34** | 16.89 | 34.50 | **86.87** | **30.94** | 3.54 | 10.00 | **40.65** | 38.22 | 26.82 |
| ReST-KV | 14.97 | **20.02** | **32.61** | **35.27** | **29.15** | 10.71 | **17.21** | 19.12 | **18.39** | **42.00** | 85.35 | 30.57 | 3.54 | **12.00** | 40.01 | **41.41** | **28.27** |
| Llama2-13B-Chat, $B_{\text{total}} = 512L$ | | | | | | | | | | | | | | | | | |
| StreamingLLM | 14.80 | 19.01 | 21.58 | 33.08 | 28.92 | 12.43 | 20.27 | 18.27 | 19.82 | 56.50 | 85.98 | 33.02 | 4.05 | 7.50 | 49.21 | 47.83 | 29.52 |
| H2O | 17.28 | 20.94 | 27.81 | 32.98 | 29.39 | 10.66 | **21.57** | 19.50 | **24.49** | 61.50 | 82.61 | 34.51 | **4.34** | 9.50 | 47.58 | 45.41 | 30.63 |
| TOVA | 16.95 | 21.97 | 33.18 | 35.51 | 31.11 | 13.94 | 20.33 | 19.74 | 22.94 | 62.33 | 85.54 | 35.06 | 2.94 | 10.57 | 49.60 | 49.80 | 31.97 |
| SnapKV | 17.46 | 22.46 | 33.79 | **36.39** | 31.37 | 14.46 | 20.51 | **20.51** | 22.76 | 62.50 | 86.42 | **35.88** | 3.55 | **11.50** | **50.12** | 50.08 | 32.48 |
| ReST-KV | **18.00** | **23.72** | **34.49** | 36.21 | **32.55** | **15.43** | 20.67 | 20.20 | 24.24 | **66.50** | **87.37** | 35.01 | 4.04 | **11.50** | 49.90 | **50.89** | **33.17** |
| Llama3-70B-Instruct, $B_{\text{total}} = Full$ | | | | | | | | | | | | | | | | | |
| Full | 27.75 | 46.48 | 49.68 | 52.04 | 54.90 | 30.44 | 32.37 | 22.20 | 27.62 | 73.50 | 92.46 | 45.72 | 12.00 | 72.50 | 41.70 | 69.06 | 46.90 |
| Llama3-70B-Instruct, $B_{\text{total}} = 64L$ | | | | | | | | | | | | | | | | | |
| StreamingLLM | 24.11 | 27.63 | 25.53 | 41.00 | 48.39 | 23.77 | 16.92 | 20.14 | 17.07 | 40.00 | 77.20 | 37.10 | 12.00 | **72.50** | 44.82 | 58.88 | 36.69 |
| H2O | 24.07 | 31.33 | 27.49 | 44.83 | 49.09 | 25.14 | **22.31** | 20.59 | **24.30** | 49.50 | **91.45** | 40.29 | 12.00 | **72.50** | 44.97 | 60.63 | 40.03 |
| TOVA | 24.53 | 30.43 | 27.56 | 45.29 | 49.64 | 25.93 | 22.30 | 20.08 | 23.46 | 48.66 | 91.18 | 40.23 | 11.85 | **72.50** | 44.31 | 60.65 | 39.91 |
| SnapKV | 23.97 | 32.92 | 34.96 | 46.35 | 52.90 | 26.05 | 18.33 | **21.55** | 19.98 | 43.00 | 88.83 | **41.18** | 12.00 | **72.50** | 44.42 | **61.63** | 40.04 |
| ReST-KV | **26.32** | **36.38** | **38.44** | **49.51** | **53.18** | **26.20** | 21.81 | 20.02 | 21.48 | **59.75** | 88.51 | 40.51 | **12.05** | 71.50 | **45.22** | 61.22 | **42.01** |
| Llama3-70B-Instruct, $B_{\text{total}} = 512L$ | | | | | | | | | | | | | | | | | |
| StreamingLLM | 24.62 | 31.89 | 31.23 | 44.91 | 47.51 | 25.91 | 23.08 | 19.76 | 24.15 | 62.50 | 88.14 | 43.36 | 12.00 | **72.50** | 48.71 | 66.04 | 41.64 |
| H2O | 27.56 | 42.91 | 36.19 | 50.40 | 49.87 | 25.98 | **28.82** | 21.67 | 27.06 | 72.00 | 91.88 | 44.57 | 12.00 | 72.00 | 42.65 | 67.87 | 44.59 |
| TOVA | 27.51 | 42.49 | 35.71 | 51.02 | 50.42 | 25.12 | 27.88 | 21.60 | **27.28** | 72.04 | 92.04 | 45.13 | **12.89** | 71.18 | 43.51 | 68.53 | 44.65 |
| SnapKV | 27.67 | 44.58 | 48.00 | **51.66** | 53.73 | **30.61** | 24.80 | **22.82** | 25.89 | 70.00 | **92.63** | 45.14 | 12.00 | **72.50** | 44.59 | **69.20** | 45.99 |
| ReST-KV | **27.85** | **45.21** | **50.06** | 51.55 | **54.45** | 29.83 | 25.77 | 22.54 | 25.83 | **72.50** | **92.63** | **46.59** | 12.00 | **72.50** | 43.44 | 68.95 | **46.36** |

tains a clear and consistent margin over the baselines, demonstrating its robustness in extended-context scenarios.

Table 12: LongBench comparison with LaCache on Llama2-7B-Chat at 25% cache budget.

| Method | NrtvQA | Qasper | MF-en | HotpotQA | 2WikiMQA | Musique | GovRpt | QASum | MultiNews | TREC | TriviaQA | SAMSum | PCount | PRe | Lcc | RB-P | Avg |
|---|---|---|---|---|---|---|---|---|---|---|---|---|---|---|---|---|---|
| FullKV | 18.39 | 20.11 | 35.67 | 31.25 | 25.50 | 10.14 | 25.68 | 20.93 | 26.27 | 64.00 | 83.38 | 40.99 | 5.50 | 10.00 | 60.81 | 55.27 | 33.37 |
| StreamingLLM | 15.07 | 17.68 | 24.69 | 29.15 | 22.66 | 6.90 | 21.16 | 20.34 | 24.76 | 59.50 | 81.68 | 39.17 | 6.00 | 4.00 | 57.51 | 54.74 | 30.31 |
| LaCache | 15.27 | 17.68 | 24.69 | 29.15 | 22.66 | 6.90 | 21.56 | 20.54 | 24.76 | 59.70 | 81.68 | 39.57 | 6.00 | 4.00 | 57.71 | 55.14 | 30.44 |
| **ReST-KV** | **18.36** | **19.00** | **34.44** | **31.51** | **25.16** | **10.61** | **21.85** | **20.78** | **23.77** | **60.90** | **83.63** | **40.32** | **5.82** | **10.20** | **59.53** | **54.69** | **32.54** |

Table 13: LongBench comparison with LaCache on Llama2-7B-Chat at 50% cache budget.

| Method | NrtvQA | Qasper | MF-en | HotpotQA | 2WikiMQA | Musique | GovRpt | QASum | MultiNews | TREC | TriviaQA | SAMSum | PCount | PRe | Lcc | RB-P | Avg |
|---|---|---|---|---|---|---|---|---|---|---|---|---|---|---|---|---|---|
| FullKV | 18.39 | 20.11 | 35.67 | 31.25 | 25.50 | 10.14 | 25.68 | 20.93 | 26.27 | 64.00 | 83.38 | 40.99 | 5.50 | 10.00 | 60.81 | 55.27 | 33.37 |
| StreamingLLM | 17.58 | 18.57 | 25.97 | 27.39 | 22.42 | 9.80 | 20.97 | 19.76 | 24.51 | 63.00 | 82.50 | 40.68 | 7.00 | 4.00 | 61.04 | 54.36 | 31.22 |
| LaCache | 18.38 | 19.57 | 26.77 | 28.19 | 23.22 | 10.80 | 21.97 | 20.96 | 25.71 | 64.00 | 83.70 | 41.68 | 7.80 | 5.20 | 61.84 | 55.16 | 32.18 |
| **ReST-KV** | **18.04** | **19.41** | **35.77** | **30.92** | **25.33** | **10.36** | **24.04** | **21.50** | **25.22** | **62.20** | **83.63** | **40.56** | **5.94** | **10.05** | **60.90** | **54.56** | **33.03** |

A breakdown by task category reveals that ReST-KV performs particularly well on retrieval tasks (S-NIAH, MQ-NIAH, MV-NIAH) and multi-hop reasoning (VT), often approaching the accuracy levels of the full KV cache. These results indicate that ReST-KV is effective at preserving semantically salient tokens under constrained memory. More challenging tasks, such as MK-NIAH-3 and the CWE aggregation task with uniform word distributions, remain difficult across all methods. Nonetheless, ReST-KV continues to outperform other eviction baselines in these settings, suggesting stronger resilience to task complexity and noise.

# F ADDITIONAL EXPERIMENTS ON NEEDLE-IN-A-HAYSTACK TEST

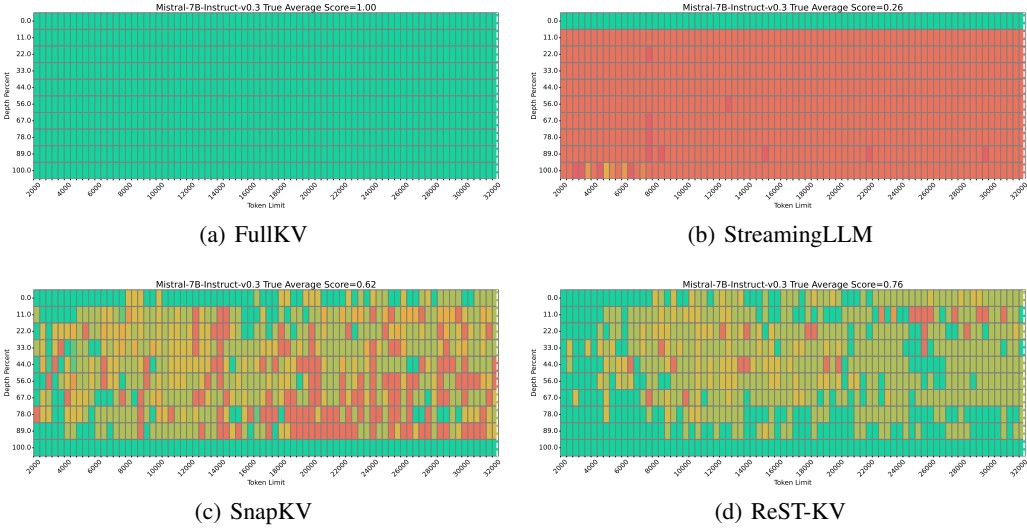

(a) FullKV      (b) StreamingLLM

(c) SnapKV      (d) ReST-KV

Figure 8: Performance comparison on the Needle in a Haystack Test using Mistral-7B-Instruct-v0.3 with $B_{\text{total}} = 128L$.

In this section, we present additional experiments to further evaluate the effectiveness of our method on the Needle-in-a-Haystack test. This benchmark assesses a model's ability to retrieve critical information embedded within long contexts. While Section 4.4 already provides results for Mistral-7B-Instruct-v0.3 with a cache budget of $B = 1024L$, we extend our analysis by considering additional settings: (1) Mistral-7B-Instruct-v0.3 with a reduced cache budget of $B = 128L$, (2) Llama3.1-8B-Instruct under both $B = 128L$ and $B = 1024L$.

Figures 8, 9 and 10 illustrate the performance comparison under these settings. We observe the following key insights:

Table 14: Performance comparison of ReST-KV and baseline eviction strategies on the RULER benchmark across multiple context lengths (4k to 128k tokens). Results are reported as average accuracy (%) over subtasks. ReST-KV consistently outperforms other methods, particularly on retrieval and multi-hop reasoning tasks. The best result is highlighted in **bold**, and the second-best is underlined.

| Method | S-NIAH-1 | S-NIAH-2 | S-NIAH-3 | MK-NIAH-1 | MK-NIAH-2 | MK-NIAH-3 | MQ-NIAH | MV-NIAH | CWE | FWE | VT | Avg. Acc |
|---|---|---|---|---|---|---|---|---|---|---|---|---|
| | | | | Llama-3.1-8B-Instruct, $B_{\text{total}} = 1024L$, context length=4k | | | | | | | | |
| Full | 100.0 | 100.0 | 99.60 | 100.0 | 100.0 | 99.60 | 99.90 | 96.25 | 99.78 | 97.67 | 99.96 | 99.34 |
| StreamingLLM | 27.8 | 30.6 | 31.40 | 34.2 | 26.2 | 29.6 | 28.7 | 30.05 | 73.2 | **96.13** | 30.0 | 39.81 |
| SnapKV | **100.00** | 99.0 | 20.6 | 99.6 | 93.0 | 31.00 | 99.2 | 90.45 | 91.46 | 95.33 | **99.96** | 83.60 |
| PyramidKV | **100.00** | 99.80 | 9.6 | **100.00** | 96.80 | 26.2 | 99.70 | 93.60 | 74.88 | 94.33 | 99.92 | 81.35 |
| ReST-KV | **100.00** | **100.00** | 99.60 | **100.00** | **100.00** | 49.80 | **99.95** | 97.00 | 91.78 | 96.07 | 99.92 | **94.01** |
| | | | | Llama-3.1-8B-Instruct, $B_{\text{total}} = 1024L$, context length=8k | | | | | | | | |
| Full | 100.0 | 100.0 | 100.0 | 100.0 | 99.80 | 98.80 | 100.0 | 95.75 | 97.62 | 95.27 | 99.92 | 98.83 |
| StreamingLLM | 11.0 | 12.0 | 13.00 | 14.2 | 11.0 | 11.80 | 12.35 | 12.45 | 4.36 | 86.93 | 13.56 | 18.42 |
| SnapKV | **100.00** | 98.4 | 12.0 | 98.0 | 85.6 | 7.6 | 97.8 | 87.45 | 56.06 | 88.27 | 99.76 | 75.54 |
| PyramidKV | **100.00** | 99.80 | 3.0 | 99.20 | 87.80 | 5.2 | 98.05 | 88.25 | 40.24 | 89.00 | 99.68 | 73.66 |
| ReST-KV | **100.00** | **100.00** | 95.60 | **100.00** | 99.80 | 19.60 | **100.00** | 95.80 | 51.16 | 91.53 | 99.76 | **86.66** |
| | | | | Llama-3.1-8B-Instruct, $B_{\text{total}} = 1024L$, context length=16k | | | | | | | | |
| Full | 100.0 | 100.0 | 100.0 | 99.60 | 100.0 | 99.00 | 99.85 | 98.25 | 90.90 | 96.67 | 99.80 | 98.55 |
| StreamingLLM | 5.6 | 6.4 | 5.80 | 7.2 | 6.0 | 5.00 | 5.2 | 6.7 | 0.18 | 78.53 | 6.52 | 12.1 |
| SnapKV | **100.00** | 97.0 | 4.0 | 97.8 | 74.0 | 3.8 | 97.25 | 88.95 | 27.36 | 92.6 | 99.6 | 71.12 |
| PyramidKV | **100.00** | 97.40 | 1.2 | 98.00 | 75.20 | 3.4 | 97.0 | 86.5 | 18.98 | 95.07 | 99.80 | 70.23 |
| ReST-KV | **100.00** | **100.00** | 93.00 | 99.60 | 99.80 | 17.00 | **100.00** | 96.10 | 22.86 | 97.13 | 99.80 | **84.12** |
| | | | | Llama-3.1-8B-Instruct, $B_{\text{total}} = 1024L$, context length=32k | | | | | | | | |
| Full | 100.0 | 100.0 | 100.0 | 100.0 | 100.0 | 99.60 | 99.90 | 98.95 | 48.60 | 97.07 | 99.68 | 94.89 |
| StreamingLLM | 3.6 | 1.8 | 2.4 | 3.0 | 3.8 | 2.00 | 2.5 | 2.45 | 0.12 | 91.13 | 3.52 | 10.57 |
| SnapKV | **100.00** | 97.20 | 6.20 | 99.40 | 61.00 | 2.0 | 96.5 | 87.25 | 16.96 | 71.27 | 98.64 | 66.95 |
| PyramidKV | **100.00** | 97.2 | 2.8 | 99.2 | 59.2 | 1.8 | 96.55 | 85.2 | 10.52 | 75.2 | 98.60 | 66.02 |
| ReST-KV | **100.00** | **100.00** | 98.20 | 99.60 | 99.00 | 15.20 | **99.95** | 97.60 | 9.36 | 83.53 | 98.16 | **81.87** |
| | | | | Llama-3.1-8B-Instruct, $B_{\text{total}} = 1024L$, context length=64k | | | | | | | | |
| Full | 100.0 | 100.0 | 99.80 | 99.80 | 99.20 | 94.00 | 99.75 | 98.95 | 7.96 | 90.60 | 98.32 | 89.85 |
| StreamingLLM | 2.0 | 1.6 | 2.4 | 2.6 | 2.0 | 0.80 | 2.05 | 2.8 | 0.14 | **90.87** | 1.76 | 9.91 |
| SnapKV | **100.00** | 96.4 | 3.20 | 99.00 | 32.2 | 0.2 | 91.7 | 58.45 | **2.86** | 54.53 | 93.60 | 57.47 |
| PyramidKV | **100.00** | 96.80 | 1.0 | 99.0 | 36.80 | 0.2 | 92.10 | 58.25 | 1.66 | 55.87 | 94.56 | 57.84 |
| ReST-KV | **100.00** | **100.00** | 90.80 | **100.00** | 96.80 | 15.60 | 98.95 | 97.30 | 1.3 | 71.67 | 92.68 | **78.65** |
| | | | | Llama-3.1-8B-Instruct, $B_{\text{total}} = 1024L$, context length=128k | | | | | | | | |
| Full | 97.40 | 97.80 | 95.20 | 96.20 | 87.00 | 63.20 | 95.85 | 94.95 | 0.06 | 64.73 | 80.08 | 79.32 |
| StreamingLLM | 0.4 | 2.0 | 3.00 | 2.4 | 0.6 | 0.80 | 1.95 | 2.35 | **1.26** | **74.73** | 0.52 | 8.18 |
| SnapKV | **97.40** | 96.80 | 1.4 | 93.8 | 25.6 | 0.0 | 80.6 | 27.0 | 0.08 | 30.47 | 74.76 | 47.99 |
| PyramidKV | **97.40** | 96.8 | 0.2 | 94.20 | 30.80 | 0.4 | 80.95 | 29.20 | 0.14 | 32.07 | 76.08 | 48.93 |
| ReST-KV | **97.40** | 98.00 | 75.80 | 95.40 | 74.00 | 3.60 | 92.15 | 93.70 | 0.16 | 47.73 | 73.12 | **68.28** |

(a) FullKV    (b) StreamingLLM    (c) SnapKV    (d) ReST-KV

Figure 9: Performance comparison on the Needle in a Haystack Test using Llama3.1-8B-Instruct with $B_{\text{total}} = 128L$.

- **Mistral-7B-Instruct-v0.3** ($B = 128L$) retains **76%** of the original accuracy, outperforming SnapKV by **14%**. This demonstrates that our method maintains strong retrieval capability even under severe cache constraints.

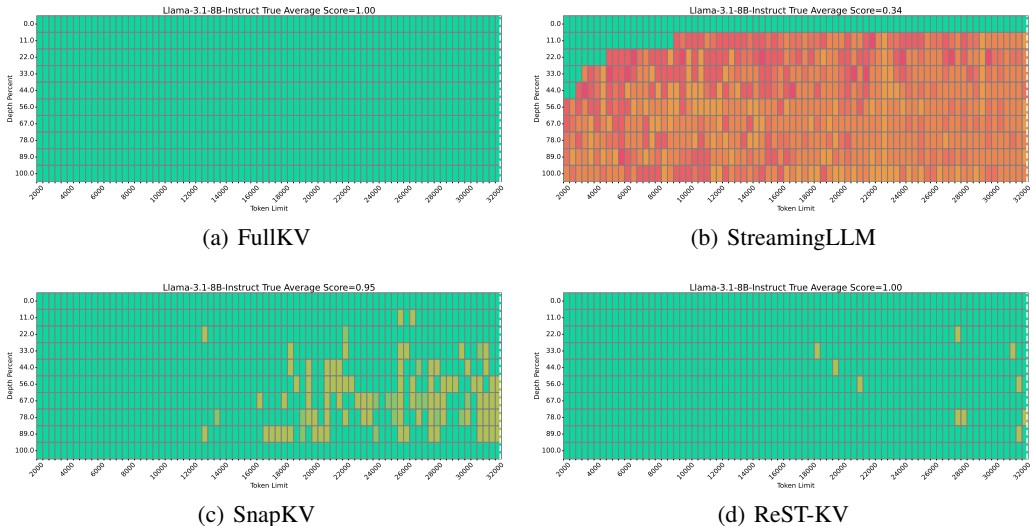

Figure 10: Performance comparison on the Needle in a Haystack Test using Llama3.1-8B-Instruct with $B_{\text{total}} = 1024L$.

- **Llama3.1-8B-Instruct** ($B = 128L$) achieves **74%** accuracy, surpassing SnapKV by **6%**, indicating its robustness in preserving key-value pairs under limited cache budgets.
- **Llama3.1-8B-Instruct** ($B = 1024L$) attains **100%** accuracy, meaning it can match full KV cache performance while storing only **1/32** of the original tokens. This highlights the efficiency of our approach in long-context retrieval with minimal memory usage.

These results further validate the robustness and efficiency of our method in selecting the most relevant KV pairs while minimizing memory overhead. Notably, even with a significantly reduced cache budget, our approach consistently outperforms prior methods, ensuring reliable long-context retrieval across different models and settings.

## G    ADDITIONAL EXPERIMENTS ON ABLATION STUDY

In this section, we conduct additional ablation experiments to rigorously analyze the effectiveness of core components in ReST-KV and assess its sensitivity to key hyper-parameters.

### G.1    EFFICACY OF THE PROPOSED OUTPUT RECONSTRUCTION INDICATOR

To evaluate the proposed eviction indicator, we compare it with different types of eviction indicators under the same baseline, including random selection, attention weights, attention weights weighted by the values's norm ($\mathbf{A}_t[n] \cdot \|\mathbf{v}_n\|_2$), similar to the VATP method (Guo et al., 2024), and our output reconstruction. As shown in Table 15, directly weighting attention weights by the values's norm does not effectively incorporate the values information. Our method significantly outperforms all baselines, indicating that the layer-wise output reconstruction perspective better assesses the importance of KV cache.

### G.2    EFFICACY OF THE PROPOSED SPATIAL-TEMPORAL SMOOTHING

To assess the effectiveness of the spatial-temporal smoothing mechanism, we perform an ablation study to examine the impact of different smoothing methods. As shown in the left part of Table 16, various temporal smoothing techniques, including Mean, Inv-EMA, and EMA, are tested. Notably, EMA smoothing achieves the best performance, surpassing other baselines, which demonstrates its effectiveness in capturing temporal variations by giving higher weights to more recent KV pairs.

Table 15: Ablation study on different types of information considered by the eviction indicator. Using output reconstruction as the eviction criterion achieves the best performance, surpassing methods based on attention weights or their combinations.

| Information Considered by Eviction Indicator | Avg. |
|---|---|
| Random | $6.83 \pm 0.20$ |
| Attention weights (SnapKV) | 33.95 |
| Attention weights and values (VATP) | 33.88 |
| Output reconstruction (Eq. equation 7) | **35.86** |

Table 16: Ablation study on the effect of different temporal and spatial smoothing methods in the eviction indicator. EMA refers to our proposed exponential moving average temporal smoothing, while AWS represents our adaptive window-based spatial smoothing.

| Temporal Smoothing | Avg. | Spatial Smoothing | Avg. |
|---|---|---|---|
| None | 35.22 | None | 33.50 |
| Mean | 34.02 | Avgpool | 35.69 |
| Inv-EMA | 31.25 | Maxpool | 35.59 |
| EMA (Ours) | **35.86** | AWS (Ours) | **35.86** |

In addition, we evaluate the spatial smoothing methods, as detailed in the right part of Table 16. Methods such as Avgpool, Maxpool, and our adaptive window-based smoothing (AWS) are compared, with AWS achieving the highest average performance. This suggests that the adaptive window-based approach, significantly enhances the eviction indicators ability to adjust for varying window sizes and offsets, thereby improving the assessment of the importance of KV pairs in the spatial-temporal context.

### G.3 HYPER-PARAMETER SENSITIVITY ANALYSIS

The sensitivity analysis of the temporal smoothing factor $\alpha$ and spatial smoothing scaling factor $\beta$ is presented in Figure 7 in the main text (Section 4.5). Both hyperparameters exhibit stable performance across wide ranges, and all experiments adopt the fixed setting $\alpha = 0.3$, $\beta = 2000$ without per-model or per-task tuning.

## H ADDITIONAL EXPERIMENTS ON EFFICIENCY

In this section, we investigate the integration of ReST-KV with prefill optimization techniquesexemplified by Minference (Jiang et al., 2024) and FlexPrefill to assess potential improvements in Time To First Token (TTFT). To this end, we conduct additional experiments on the RULER 128k benchmark using the `LLaMA3.1-8B-Instruct` model, focusing on the efficiency of our proposed KV cache eviction method, particularly its impact on TTFT and decoding latency. Results are summarized in Table 17.

Table 17: Efficiency analysis on RULER 128k. All results are normalized to the Full KV caching baseline.

| Method | 128k Avg. Acc. | TTFT | Decoding Latency |
|---|---|---|---|
| Full | 79.32 | $1\times$ | $1\times$ |
| ReST-KV | 68.28 | $0.97\times$ | $10.61\times$ |
| ReST-KV+MInference | 53.71 | $2.99\times$ | $10.41\times$ |
| ReST-KV+FlexPrefill($\gamma = 0.9$) | 67.16 | $3.42\times$ | $10.46\times$ |
| ReST-KV+FlexPrefill($\gamma = 0.95$) | 68.12 | $2.37\times$ | $10.54\times$ |

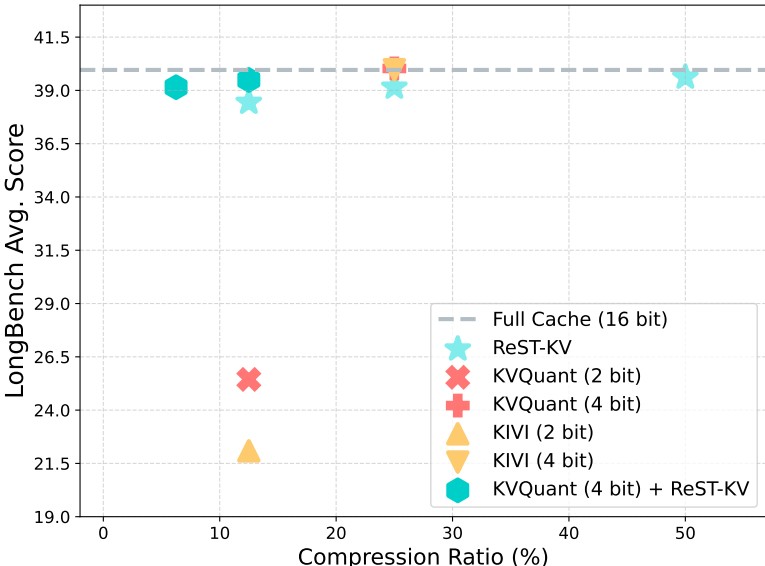

Figure 11: Comparison of ReST-KV, KV cache quantization methods (KIVI and KVQuant), and their combination on Llama3.1-8B-Instruct using LongBench dataset.

Our method is a KV cache eviction strategy that achieves a substantial improvement in decoding latencyover $10\times$ speedupwhile maintaining a comparable TTFT (0.97Œ) to full KV caching. Importantly, it maintains a high level of accuracy (68.28%), demonstrating that our eviction strategy preserves model performance effectively even under long context scenarios.

Furthermore, our method is orthogonal and compatible with sparse prefilling techniques such as MInference (Jiang et al., 2024) and FlexPrefill (Lai et al., 2025). When combined with these methods, we observe additional gains in TTFT. For example, integrating FlexPrefill with $\gamma = 0.95$ achieves a 2.37Œ TTFT speedup while retaining high decoding efficiency (10.54Œ latency speedup) and competitive accuracy (68.12%). This shows that our approach not only accelerates decoding but also enables efficient and flexible integration with other prefill optimization techniques.

## I  INTEGRATION WITH KV CACHE QUANTIZATION

In this section, we further investigate the interplay between ReST-KV and established KV cache quantization techniques, specifically KIVI (Liu et al., 2024b) and KVQuant (Hooper et al., 2024). Our goal is to evaluate whether combining ReST-KVa KV eviction method that accounts for the effects of attention redistribution and the spatial-temporal dynamics in KV selection can synergize with quantization or even outperform aggressive quantization applied to a full, non-evicted KV cache under similar overall compression ratios.

To this end, we compare ReST-KV, both in isolation and combined with KIVI and KVQuant, against a baseline using full KV cache with various bit-width quantizations. Figure 11 visually summarizes the results.

In particular, even with a stringent total compression ratio of 6.25%, achieved by combining ReST-KV with moderate 4-bit quantization, ReST-KV retains high average accuracy. In contrast, applying more aggressive 2-bit KIVI or KVQuant directly to the full KV cache results in significantly lower accuracy.

These results suggest that eviction strategies which explicitly account for attention redistribution and spatial-temporal token redundancy can provide a more effective pathway to KV cache compression than quantization-only approaches. The combination of ReST-KV and lightweight quantization thus offers a practical and robust solution for efficient inference under tight memory constraints.

# J ADDITIONAL EXPERIMENTS ON INFINITEBENCH

In this section, we evaluate ReST-KV on the InfiniteBench benchmark (Zhang et al., 2024b) to further assess its long-context capabilities. InfiniteBench tests LLM performance on extremely long sequences through a diverse set of tasks. These tasks include realistic scenarios such as novel-based reasoning (summarization, QA, multiple-choice, using novels with key entity replacement), dialogue understanding, and code debugging. Additionally, synthetic tests probe specific long-context abilities like retrieval, state preservation, and sequential processing.

Experiments are conducted on the Llama3.1 model. We compare ReST-KV against SnapKV (Li et al., 2024b), as both are post-prefill KV eviction strategies. To ensure a direct comparison of their eviction effectiveness, both methods retain a fixed KV cache budget of 1024 tokens post-eviction, regardless of the initial input context length.

Table 18 details the average performance across InfiniteBench subtasks. ReST-KV achieves a notably higher overall average accuracy than SnapKV (e.g., 38.8% vs. 36.8%). This performance advantage is particularly evident in retrieval-focused tasks (Retrieve.PassKey, Retrieve.Number, Retrieve.KV), where SnapKV can exhibit critical failures on some subtasks. ReST-KV also generally demonstrates stronger results in question answering (En.QA, Zh.QA) and Math.Find. While SnapKV may be competitive on select tasks like En.Sum, the consistent and superior performance of ReST-KV across a wider range of demanding retrieval and reasoning tasks contributes to its substantially higher overall average. These findings underscore the efficacy of ReST-KV's reconstruction-aware eviction strategy when applied to the challenging long-context scenarios presented by InfiniteBench. Notably, the En.QA subset has an average input length of **192.6k** tokens and the Zh.QA subset averages **2,068.6k** tokens (Zhang et al., 2024b), directly covering the $\geq$256k regime that extends well beyond the 128k setting of the main experiments. The consistent gains of ReST-KV across these multi-hundred-thousand to multi-million token sequences confirm that neither the fixed observation window nor the EMA-based temporal smoothing introduces cumulative degradation at extreme context lengths.

Table 18: Performance of different methods on InfiniteBench.

| Methods | Retr.PassKey | Retr.Num | Retr.KV | En.Dia | En.Sum | En.MC | En.QA | Zh.QA | Math.Find | Debug | Avg. |
|---------|--------------|----------|---------|--------|--------|-------|-------|-------|-----------|-------|------|
| ReST-KV | **100.0** | **93.7** | **11.4** | **10.5** | 22.9 | 67.2 | **13.2** | **13.1** | **34.0** | **22.3** | **38.8** |
| SnapKV | **100.0** | 87.1 | 0.0 | 10.0 | **23.7** | **67.7** | 11.3 | 12.2 | **34.0** | **22.3** | 36.8 |

