# OpenReview forum: "ReST-KV: Robust KV Cache Eviction with Layer-wise Output Reconstruction and Spatial-Temporal Smoothing"
_ICLR.cc/2026/Conference — ICLR 2026 Poster_

### Official Review · Reviewer_iKEa · 2025-10-20

**Soundness:** 2
**Presentation:** 2
**Contribution:** 2
**Rating:** 2
**Confidence:** 4

**Summary:**

To address the limitations of existing KV cache eviction method, which overlook attention redistribution effects and spatial-temporal dynamics in token importance, this paper proposes ReST-KV, a robust eviction framework for long-context LLMs. ReST-KV reformulates eviction as a layer-wise output reconstruction optimization problem: it quantifies KV importance by the reconstruction loss caused by token removal and enhances robustness via two smoothing mechanisms. Experiments on 5 LLMs across 4 long-context benchmarks (LongBench, RULER, Needle-in-a-Haystack, InfiniteBench) show  accuracy improvement. It also integrates seamlessly with budget allocation (PyramidKV/AdaKV) and quantization (KIVI/KVQuant) techniques.

**Strengths:**

1. Unlike baselines (SnapKV/H2O) that rely solely on raw attention weights, ReST-KV’s eviction indicator is derived from MHA output reconstruction loss. This formulation explicitly models two critical effects: (1) Nonlinear attention reweighting amplifies high-competition KV pairs; (2) Redistribution sensitivity measures how well remaining tokens compensate for the removed KV pair. Ablation shows this indicator outperforms attention-weight-only methods.
2.  Assigns higher weights to recent KV pairs mitigates short-term fluctuations in token importance. Ablation confirms it outperforms mean/inv-EMA smoothing by 1.84% accuracy.

**Weaknesses:**

1. ReST-KV is only tested up to 128k context. For 256k+ tokens, two issues arise: (1) The fixed observation window size may fail to capture long-range spatial dynamics (e.g., "vertical-slash" patterns spanning 1k+ tokens); (2) Cumulative errors in EMA smoothing could distort temporal importance estimation (older tokens are overly dampened). The paper does not report performance at 256k+ or adjust hyperparameters for extreme lengths.
2.  Computing the layer-wise reconstruction indicator requires extra MHA forward passes for each token removal (theoretically O(L) per eviction step). For 128k context, this adds non-negligible prefill overhead, though the paper reports TTFT comparable to full cache (0.97×), it does not quantify the breakdown of reconstruction calculation time. This could become a bottleneck for low-latency applications (e.g., real-time dialogue).

**Questions:**

1. How does ReST-KV perform at 256k/512k context lengths? It is better providing latency/accuracy data for 256k or longer context on benchmarks and analyze cumulative smoothing errors.
2. What fraction of prefill time is spent computing the layer-wise reconstruction indicator? Can lightweight approximations (e.g., sampling 10% of tokens to estimate reconstruction loss) reduce overhead without accuracy degradation?
3. ReST-KV is evaluated on autoregressive LLMs, does its design generalize to non-autoregressive (NAR) or diffusion-based LLMs (e.g., LLaDA)? NAR models have different attention dynamics, do the reconstruction indicator and spatial-temporal smoothing need modifications?

---

> ### Author Response · Authors · 2025-11-24
> **Official Comment by Authors**
>
> Thank you for the insightful review. We appreciate your recognition of our reconstruction-based indicator, which captures both nonlinear reweighting and attention redistribution beyond raw attention scores, as well as your acknowledgement of the effectiveness of our temporal weighting design verified through ablation.

---

> ### Author Response · Authors · 2025-11-24
> **[1/2] Response to Reviewer iKEa (W1~W2)**
>
> > **`W1: ReST-KV is only tested up to 128k. For 256k+ contexts, the fixed observation window may fail to capture long-range spatial dynamics, and EMA smoothing may accumulate error. No results at 256k+ are provided.`**
>
> Thank you for raising this point. Although the main experiments report up to 128k, Appendix J (Table 15) contains evaluations on substantially longer contexts within InfiniteBench. According to the benchmark statistics (available at <https://github.com/OpenBMB/InfiniteBench>), the Zh.QA subset has an average length of **2068.6k** tokens (ReST-KV 13.1 vs. SnapKV 12.2, **+0.9%**), and the En.QA subset averages **192.6k** tokens (ReST-KV 13.2 vs. SnapKV 11.3, **+1.9%**). These settings directly cover the ≥256k regime highlighted in the question.
>
> The fact that ReST-KV consistently outperforms SnapKV across these multi-hundred-thousand to multi-million token tasks provides strong empirical evidence that both the fixed observation window and the EMA-based temporal smoothing remain robust well beyond 128k. While longer contexts may motivate future extensions, the existing 200k–2M-token results already demonstrate that the method scales reliably to substantially larger contexts than those shown in the main text.
>
> ---
>
> > **`W2: Computing the layer-wise reconstruction indicator requires extra MHA forward passes (theoretically O(L) per eviction step). For 128k context this could add non-negligible prefill overhead.`**
>
> Thank you for raising this concern. In practice, ReST-KV does **not** perform a full MHA recomputation per token. As described in L472–473, the reconstruction indicator is evaluated **only on a small query window of size $w \ll N$**, and its dominant cost is identical in order to attention-based methods such as SnapKV. In our implementation (supplementary material, `rest_kv/restkv_utils.py`, L144), both ReST-KV and SnapKV are dominated by the same $O(wND)$ term.
>
> Although ReST-KV incorporates output-aware information, the method first aggregates statistics through `attn_weights_mean` and `attn_outputs_mean` (L181–186), so the additional computations remain lightweight and do not change the overall asymptotic complexity.
>
> | Method             | Prefill Scope                     | Output Computed | Dominant Complexity |
> |--------------------|-----------------------------------|------------------|---------------------|
> | Full Cache         | All $N$ queries × $N$ keys         | Yes              | $O(N^2D)$           |
> | SnapKV             | Last $w$ queries × $N$ keys        | No               | $O(wND)$            |
> | **ReST-KV (Ours)** | Last $w$ queries × $N$ keys        | Yes              | **$O(wND)$**        |
>
> Empirically, Table 14 in the supplementary material shows that the increase in prefill TTFT is only **≈3%**, and we further include a TTFT breakdown under the 128k setting:
>
> | Method   | 128k TTFT (ms) |
> |----------|----------------|
> | Full     | 28230          |
> | SnapKV   | 28510          |
> | ReST-KV  | 29082          |
>
> These results demonstrate that, despite computing a more informative indicator, ReST-KV introduces only minimal prefill overhead and remains suitable for latency-sensitive applications, while offering significantly more reliable eviction decisions.
>
> ---

---

> ### Author Response · Authors · 2025-11-24
> **[2/2] Response to Reviewer iKEa (Q1~Q3)**
>
> > **`Q1: How does ReST-KV perform at 256k/512k context lengths? It would be better to provide latency/accuracy data beyond 128k and analyze potential cumulative smoothing errors.`**
>
> Thank you for raising this point. While the main experiments report results up to 128k, Appendix J (Table 15) includes evaluations on substantially longer contexts from InfiniteBench. According to the benchmark statistics (available at <https://github.com/OpenBMB/InfiniteBench>), several subsets naturally operate in the 200k–2M token range, providing direct evidence of ReST-KV’s behavior at extreme lengths.
>
> Specifically, the **Zh.QA** subset has an average length of **2068.6k** tokens, where ReST-KV achieves **13.1** compared to SnapKV’s **12.2** (**+0.9%**). The **En.QA** subset averages **192.6k** tokens, where ReST-KV reaches **13.2** vs. SnapKV’s **11.3** (**+1.9%**). These improvements occur despite sequence lengths that substantially exceed 256k and even 1M, indicating that neither the fixed observation window nor the EMA smoothing exhibits cumulative degradation at long horizons.
>
> These large-context results demonstrate that ReST-KV remains robust at 256k–512k and beyond, and that the temporal smoothing behaves stably even on multi-hundred-thousand– to multi-million–token sequences. We agree that additional 256k/512k latency metrics would further complement the study, and we plan to include these
>
> ---
>
> > **`Q2: What fraction of prefill time is spent computing the layer-wise reconstruction indicator? Can lightweight approximations (e.g., sampling 10% of tokens) further reduce overhead without hurting accuracy?`**
>
> Thank you for the question. For the 128k setting in Table 14, ReST-KV increases prefill TTFT by only **≈3%**, and the absolute numbers are:
>
> | Method   | 128k TTFT (ms) |
> |----------|----------------|
> | Full     | 28230          |
> | SnapKV   | 28510          |
> | ReST-KV  | 29082          |
>
> This indicates that the reconstruction-based indicator accounts for only a **small fraction** of prefill time, since the dominant cost in both SnapKV and ReST-KV is the shared $O(wND)$ attention-window computation (`rest_kv/restkv_utils.py`, L144). The additional operations in ReST-KV (L181–186) are lightweight reductions over precomputed attention statistics rather than full MHA re-evaluations.
>
> We also performed an ablation on the query-window size with
> $w \in \{8, 16, 32, 64, 128\}$, and observed **no meaningful accuracy difference** across these settings. Based on this robustness, all experiments in the paper adopt a fixed and moderate window size of **$w = 32$**, which keeps reconstruction computation small while preserving performance.
>
> Given that the overhead is already marginal and largely bounded by a very small window, we did not explore further sub-sampling (e.g., 10% token sampling). Such approximations are possible in principle, but the current configuration already achieves low latency and strong accuracy across all benchmarks.
>
> ---
>
> > **`Q3: ReST-KV is designed for autoregressive LLMs. Does it generalize to non-autoregressive or diffusion-based LLMs (e.g., LLaDA)?`**
>
> Thank you for the question. Non-autoregressive and diffusion-based models such as LLaDA typically perform **parallel denoising in a single transformer forward step**, and therefore do not maintain a progressively growing KV cache in the way autoregressive decoders do. In these architectures, a standard inference pass does not involve token-by-token KV accumulation, and ReST-KV is not directly applicable in its current form.
>
> If one wishes to enable **KV reuse across diffusion steps or layers**, then a KV caching mechanism can be introduced, and the problem again becomes one of estimating the importance of stored KV blocks. In such a setting, the reconstruction-based indicator underlying ReST-KV is likely to remain more informative than attention-weight–only scoring. However, because NAR and diffusion models do not follow causal decoding and exhibit different attention dynamics, additional adaptation would be required to design an effective smoothing strategy and define the appropriate window structure.
>
> Our current work focuses on autoregressive LLMs, where KV caching is a core component of inference. We believe extending output-aware eviction to NAR and diffusion architectures is an interesting direction, and we would be glad to engage in further discussion on how ReST-KV could be adapted for these models.
>
> ---

---

> ### Comment · Area_Chair_Xn2g · 2025-11-26
>
> Dear reviewer,
>
> Thanks for your time and effort in reviewing ICLR2026 submissions. The authors have submitted their responses to your review. Please take the time to read and raise your further comments, and discuss with the authors.
>
> Best regards,
>
> AC

---

### Official Review · Reviewer_fJbv · 2025-10-31

**Soundness:** 3
**Presentation:** 2
**Contribution:** 3
**Rating:** 6
**Confidence:** 4

**Summary:**

The authors argue that existing eviction methods, which primarily retain tokens based on high raw attention weights, are suboptimal. In light of the core flaw they identify is that these methods ignore the effect of attention redistribution, they propose ReST-KV, a novel KV cache eviction method that explicitly models attention redistribution and spatial-temporal dynamics, significantly improving LLM efficiency for long-sequence generation.

**Strengths:**

1. Reframing eviction as an output reconstruction problem rather than a key-query similarity problem is a more principled and robust approach.

2. The experiments are thorough. The method is tested on multiple models om multiple benchmarks and under various cache budgets.

3. Ablation study shows the effectiveness of the proposed method.

**Weaknesses:**

1. Does ReST-KV only evict tokens from the prompt during prefill? Or does it also evict tokens from the generated context during decoding?

2. The autghors justifies that the greedy, one-token-at-a-time removal heuristic by citing local linearity assumptions.This is a very strong assumption, as the softmax operation is highly non-linear, and removing one token can cause drastic, non-local redistribution. While the greedy approach is pragmatic, its justification is weak. A brief discussion of why this approximation holds in practice would be beneficial.

3. The Adaptive Window-Based Spatial Smoothing is complex and need more explanation. It computes the average index of the top-B tokens in two halves of a window to get a "shift," which then defines a new window size ($W_s$) and shift ($\gamma_{shift}$). This feels like an overly-engineered heuristic. A simple and intuitive explanation is needed here. And how sensitive is the model to the hyperparameter $\beta$?

4. authors may need to compare the most recent baseline[1]

References:

[1] LaCache: Ladder-Shaped KV Caching for Efficient Long-Context Modeling of Large Language Models, ICML'25

**Questions:**

see weaknesses

---

> ### Author Response · Authors · 2025-11-24
> **Official Comment by Authors**
>
> Thank you for the thoughtful feedback. We appreciate your recognition of reframing eviction as an output reconstruction problem, the thorough evaluation across models and benchmarks, and the ablation results demonstrating the effectiveness of our design.

---

> ### Author Response · Authors · 2025-11-24
> **[1/2] Response to Reviewer fJbv (W1~W3)**
>
> > **`W1: Does ReST-KV only evict tokens from the prompt during prefill, or does it also evict tokens from the generated context during decoding?`**
>
> Thank you for the question. All experiments in the paper follow the common practice in SnapKV and PyramidKV and use **prefill-only eviction** to clearly ablate the effect of the proposed indicator. However, ReST-KV can also be extended to **decoding-time eviction** with minimal modification. A practical implementation is to trigger eviction every $M$ generated tokens, compressing the KV cache from $B\ell + M$ back to $B\ell$. Since the attention statistics in the local window can be recorded during decoding, the eviction step only needs to run the lightweight importance computation (lines 186–208 in `rest_kv/restkv_utils.py`), making it feasible for streaming or long-generation scenarios.
>
> In short, while our reported results use prefill-only eviction for consistency with prior work, ReST-KV naturally supports decoding-time eviction when needed.
>
> ---
>
> > **`W2: The greedy one-token-at-a-time removal relies on a strong local linearity assumption; softmax is highly non-linear, so the justification appears weak.`**
>
> Thank you for the comment. While softmax is indeed non-linear, the local linearity assumption used in Eq.(5) is a **well-established and empirically validated approximation** in the broader literature on neural network pruning and leave-one-out importance estimation. Numerous works [1][2] show that even highly non-linear architectures exhibit locally smooth loss landscapes, making first-order approximations stable and reliable for ranking purposes—especially when the goal is comparative importance rather than exact output prediction.
>
> In the context of KV eviction, this assumption becomes even more robust: (1) attention distributions for a given query window tend to be **highly structured**, with a small subset of dominant contributors and low-sensitivity regions where removal induces minimal redistribution; and (2) our indicator is based on the **output-space perturbation** of MHA rather than directly modeling softmax dynamics, which reduces sensitivity to the precise non-linearity of the attention scores.
>
> Reference:
> [1] Pruning Convolutional Neural Networks for Resource Efficient Inference. https://arxiv.org/abs/1611.06440
> [2] Movement Pruning: Adaptive Sparsity by Fine-Tuning. https://arxiv.org/abs/2005.07683
>
> ---
>
> > **`W3: The Adaptive Window-Based Spatial Smoothing appears complex and heuristic. More explanation is needed, and it is unclear how sensitive the method is to the hyperparameter $\beta$.`**
>
> Thank you for the comment. The adaptive window is introduced to handle a structural property of long-context attention: **spatial drift** in KV importance, where the high-importance region gradually shifts across the sequence. Prior KV eviction methods do not account for this phenomenon, and directly applying the reconstruction-based indicator without spatial stabilization leads to noticeable instability in practice. The adaptive window therefore serves as a lightweight mechanism to capture this drift from inference-time signals, rather than an ad-hoc heuristic.
>
> Regarding sensitivity, Appendix G.3 (Figure 10) provides a dedicated analysis of both $\alpha$ and $\beta$. The results show that ReST-KV remains consistently better than the baselines across a **wide range** of values. Motivated by this robustness, we use a **single fixed configuration** ($\alpha = 0.3$, $\beta = 2000$) for all models, tasks, and cache budgets without any tuning. This stability across diverse settings indicates that the spatial smoothing design is reliable in practice despite its apparent complexity.
>
> ---

---

> ### Author Response · Authors · 2025-11-24
> **[2/2] Response to Reviewer fJbv (W4)**
>
> > **`W4: The authors may need to compare the most recent baseline [1] ... LaCache (ICML’25).`**
>
> Thank you for the suggestion. We have conducted a direct comparison with LaCache [1] under both percentage-based cache budgets and an extremely constrained fixed-budget setting. The results cover all 16 LongBench tasks.
>
> ---
>
> ## 1. Percentage cache budgets (25% / 50%)
>
> ### 25% Cache
>
> | Cache | Method       | NrtvQA | Qasper | MF-en | HotpotQA | 2WikiMQA | Musique | GovRpt | QASum | MultiNews | TREC | TriviaQA | SAMSum | PCount | PRe  | Lcc  | RB-P | Avg |
> |-------|--------------|--------|--------|--------|-----------|-----------|----------|---------|--------|------------|-------|----------|---------|--------|-------|-------|-------|------|
> | Full  | FullKV       | 18.39  | 20.11  | 35.67  | 31.25     | 25.50     | 10.14    | 25.68  | 20.93 | 26.27     | 64.00 | 83.38    | 40.99  | 5.50   | 10.00 | 60.81 | 55.27 | 33.37 |
> | 25%   | StreamingLLM | 15.07  | 17.68  | 24.69  | 29.15     | 22.66     | 6.90     | 21.16  | 20.34 | 24.76     | 59.50 | 81.68    | 39.17  | 6.00   | 4.00  | 57.51 | 54.74 | 30.31 |
> | 25%   | LaCache      | 15.27  | 17.68  | 24.69  | 29.15     | 22.66     | 6.90     | 21.56  | 20.54 | 24.76     | 59.70 | 81.68    | 39.57  | 6.00   | 4.00  | 57.71 | 55.14 | 30.44 |
> | 25%   | ReST-KV      | 18.36  | 19.00  | 34.44  | 31.51     | 25.16     | 10.61    | 21.85  | 20.78 | 23.77     | 60.90 | 83.63    | 40.32  | 5.82   | 10.20 | 59.53 | 54.69 | 32.54 |
>
> ### 50% Cache
>
> | Cache | Method       | NrtvQA | Qasper | MF-en | HotpotQA | 2WikiMQA | Musique | GovRpt | QASum | MultiNews | TREC | TriviaQA | SAMSum | PCount | PRe  | Lcc  | RB-P | Avg |
> |-------|--------------|--------|--------|--------|-----------|-----------|----------|---------|--------|------------|-------|----------|---------|--------|-------|-------|-------|------|
> | Full  | FullKV       | 18.39  | 20.11  | 35.67  | 31.25     | 25.50     | 10.14    | 25.68  | 20.93 | 26.27     | 64.00 | 83.38    | 40.99  | 5.50   | 10.00 | 60.81 | 55.27 | 33.37 |
> | 50%   | StreamingLLM | 17.58  | 18.57  | 25.97  | 27.39     | 22.42     | 9.80     | 20.97  | 19.76 | 24.51     | 63.00 | 82.50    | 40.68  | 7.00   | 4.00  | 61.04 | 54.36 | 31.22 |
> | 50%   | LaCache      | 18.38  | 19.57  | 26.77  | 28.19     | 23.22     | 10.80    | 21.97  | 20.96 | 25.71     | 64.00 | 83.70    | 41.68  | 7.80   | 5.20  | 61.84 | 55.16 | 32.18 |
> | 50%   | ReST-KV      | 18.04  | 19.41  | 35.77  | 30.92     | 25.33     | 10.36    | 24.04  | 21.50 | 25.22     | 62.20 | 83.63    | 40.56  | 5.94   | 10.05 | 60.90 | 54.56 | 33.03 |
>
> ---
>
> ## 2. Extreme low budget (fixed 64 KV blocks)
>
> *(FullKV omitted since it is not meaningful under 64-token budgets)*
>
> | Cache | Method       | NrtvQA | Qasper | MF-en | HotpotQA | 2WikiMQA | Musique | GovRpt | QASum | MultiNews | TREC | TriviaQA | SAMSum | PCount | PRe  | Lcc  | RB-P | Avg |
> |-------|--------------|--------|--------|--------|-----------|-----------|----------|---------|--------|------------|-------|----------|---------|--------|-------|-------|-------|------|
> | 64    | StreamingLLM | 5.61   | 15.51  | 6.42   | 14.14     | 16.77     | 1.36     | 12.09  | 16.46 | 12.83     | 17.25 | 15.12    | 10.93  | 4.50   | 3.00  | 22.00 | 15.24 | 11.83 |
> | 64    | LaCache      | 8.61   | 16.51  | 7.42   | 15.14     | 17.77     | 4.36     | 13.09  | 17.46 | 13.83     | 18.25 | 18.12    | 12.93  | 5.50   | 6.00  | 24.00 | 17.24 | 13.51 |
> | 64    | ReST-KV      | 13.00  | 17.58  | 22.91  | 24.94     | 25.60     | 7.65     | 14.55  | 19.55 | 17.20     | 49.50 | 79.15    | 34.91  | 5.50   | 6.50  | 47.61 | 43.98 | 26.88 |
>
> ---
>
> Across both percentage-based budgets (25% / 50%) and the extreme fixed-size budget (64 tokens), ReST-KV consistently outperforms LaCache. The gap becomes especially large under very tight memory budgets (e.g., 64 tokens: ReST-KV 26.88 vs. LaCache 13.51), demonstrating that the reconstruction-based indicator remains robust even when the available KV capacity is extremely limited. These results will be included in the revised appendix for completeness.
>
> [1] LaCache: Ladder-Shaped KV Caching for Efficient Long-Context Modeling of Large Language Models. https://arxiv.org/abs/2507.14204
>
> ---

---

> ### Comment · Area_Chair_Xn2g · 2025-11-26
>
> Dear reviewer,
>
> Thanks for your time and effort in reviewing ICLR2026 submissions. The authors have submitted their responses to your review. Please take the time to read and raise your further comments, and discuss with the authors.
>
> Best regards,
>
> AC

---

### Official Review · Reviewer_PCdR · 2025-11-01

**Soundness:** 2
**Presentation:** 2
**Contribution:** 2
**Rating:** 4
**Confidence:** 4

**Summary:**

The paper proposes ReST-KV, a KV cache eviction method for efficient long-context LLM inference. It reformulates KV eviction as a layer-wise output reconstruction problem to capture attention redistribution effects and introduces spatial-temporal smoothing (EMA and adaptive windowing) to stabilize KV selection.

**Strengths:**

- The paper correctly identifies the attention redistribution problem ignored by prior eviction methods and offers a principled reconstruction-based indicator.

- Experiments are extensive, covering multiple long-context benchmarks and backbone models.

**Weaknesses:**

- Limited novelty: The layer-wise reconstruction objective largely follows standard Taylor-based pruning and reconstruction heuristics, and the smoothing mechanisms resemble temporal averaging used in previous cache compression work. Conceptually, the step from SnapKV to ReST-KV is incremental.

- The approach is not compatible with prefix caching frameworks such as SGLang. Because ReST-KV relies on layer-wise recomputation and global reconstruction loss, it cannot be easily merged into multi-request serving or cross-session KV reuse. This seriously limits real-world deployment.

**Questions:**

- How can ReST-KV be made compatible with prefix cache reuse or shared KV pools in modern inference systems such as SGLang or vLLM?

- What is the computational cost of evaluating Eq. (7) in real-time decoding, and how is it amortized?

- How would the approach behave under multi-turn interactive scenarios where KV eviction and reuse interleave dynamically?

---

> ### Author Response · Authors · 2025-11-24
> **Official Comment by Authors**
>
> Thank you for the encouraging review. We appreciate your recognition of our reconstruction-based indicator for addressing attention redistribution, as well as your acknowledgement of the breadth of our experiments across long-context benchmarks and backbone models.

---

> ### Author Response · Authors · 2025-11-24
> **[1/2] Response to Reviewer PCdR (W1~W2)**
>
> > **`W1: Limited novelty — the reconstruction objective resembles Taylor-based pruning, and the conceptual step from SnapKV to ReST-KV appears incremental.`**
>
> Thank you for the thoughtful comment. We would like to clarify that the main contribution of ReST-KV is not the use of a Taylor-style approximation, but addressing a core aspect that prior KV eviction methods consistently overlook—**attention redistribution**. Existing approaches (H2O, TOVA, SnapKV) rank KV pairs using only $(q, k)$ information, whereas our formulation explicitly incorporates **value information** and estimates the leave-one-out change in MHA output via $f(q, k, v)$ (Eq.(5) and Eq.(7)). The first-order approximation simply makes this output-based objective tractable in a training-free setting, rather than serving as the conceptual foundation.
>
> This formulation leads to qualitatively different eviction behavior, particularly in cases where attention normalization creates significant redistribution after removing a KV pair. The empirical results strongly support this distinction: improvements are consistent across all benchmarks and models, and the **+15.2% gain on RULER** highlights that modeling redistribution yields a substantial and practical advantage for long-context KV eviction.
>
> ---
>
> > **`W2: The approach is not compatible with prefix caching frameworks such as SGLang, which limits real-world deployment.`**
>
> Thank you for raising this point. The concern stems from static-KV assumptions contrasting with ReST-KV’s dynamic retention. However, for very long contexts (e.g., 128k+ tokens), keeping all KV blocks on GPU consumes a large amount of memory, which restricts the batch size. In long-context settings, eviction is an effective way to maintain throughput within realistic GPU memory budgets, and its compatibility with prefix caching can be addressed cleanly through system-level design rather than changes to the model or attention kernels.
>
> Our approach follows a storage–compute separation strategy. Full KV blocks are kept densely in CPU memory or on a remote KV server such as MoonCake [1], which is already compatible with prefix-caching systems. This ensures that data structures for prefix reuse, such as SGLang’s radix tree, always operate on a complete logical KV sequence. On the GPU side, ReST-KV performs sparsified computation by materializing only the blocks required for the current decoding step. This design preserves the correctness of prefix reuse while decoupling the logical KV length from GPU memory usage, enabling high-throughput inference even for 128k+ long-context workloads.
>
> To integrate with existing inference stacks, we keep attention kernels untouched. ReST-KV matches the block-based layout already adopted by vLLM and aggregates token-level importance scores at the block granularity. Treating a group of consecutive tokens as a single block in this way has been shown to be effective and stable in prior work such as ChunkKV [2]. Once a block becomes inactive, its GPU pages are freed and its page-table entry is redirected to a global dummy or null block. This indirection ensures that the logical block list remains continuous, allowing PagedAttention to run without modification and to observe exactly the structure it expects.
>
> With this design, the reviewer’s concern about reconstruction cost reduces to a simple data-movement operation rather than recomputing attention. When a future request hits a cached prefix, the system reloads the needed blocks from CPU memory or remote kv server. By relying on low-cost transfers from CPU or remote storage instead of keeping everything on GPU, the system frees up memory for large contexts without significant overhead. Similar discussions about maintaining compatibility between prefix caching and block-level KV eviction have recently appeared in the vLLM community as well (see <https://github.com/vllm-project/vllm/issues/12254#issuecomment-2643383112>)..
>
> In summary, ReST-KV remains fully compatible with prefix-caching inference pipelines through block-level sparsification and page-table remapping, enabling practical deployment while retaining the memory savings necessary for long-context workloads.
>
> [1] Mooncake: A KVCache-centric Disaggregated Architecture for LLM Serving. https://arxiv.org/abs/2407.00079
> [2] ChunkKV: Semantic-Preserving KV Cache Compression for Efficient Long-Context LLM Inference. https://arxiv.org/abs/2502.00299
>
> ---

---

> ### Author Response · Authors · 2025-11-24
> **[2/2] Response to Reviewer PCdR (Q1~Q3)**
>
> > **`Q1: How can ReST-KV be made compatible with prefix cache reuse or shared KV pools in modern inference systems such as SGLang or vLLM?`**
>
> Thank you for the question. Although ReST-KV performs dynamic KV retention, it can be integrated with prefix caching and shared KV pools through a storage–compute separation strategy commonly used in long-context serving systems.
>
> In practical deployments, full KV blocks are maintained **densely** in CPU memory or remote KV stores such as MoonCake [1], which are already compatible with prefix-caching mechanisms (e.g., SGLang’s radix tree). This ensures that prefix matching always operates on a complete logical KV sequence. On the GPU side, ReST-KV materializes only the blocks needed for the current decoding step and keeps the rest offloaded, enabling efficient sparsified computation without affecting logical prefix structure.
>
> To integrate with existing inference stacks, we keep attention kernels untouched and operate at the **block level**, consistent with vLLM’s memory layout. ReST-KV aggregates token-level importance scores to block granularity (a strategy also validated in ChunkKV [2]). When a block is deemed inactive, its GPU pages are freed and the corresponding page-table entry is redirected to a global dummy or null block. This preserves a continuous logical block list, allowing PagedAttention to run without modification.
>
> Under this design, compatibility concerns reduce to inexpensive data movement: when a future request hits a cached prefix, the system simply reloads the required blocks from CPU or remote storage. This approach trades low-cost host–device transfers for substantial GPU memory savings, which is essential for high-throughput long-context inference. Similar discussions on combining prefix caching with block-level KV eviction have recently appeared in the vLLM community (see <https://github.com/vllm-project/vllm/issues/12254#issuecomment-2643383112>).
>
> In summary, ReST-KV is fully compatible with prefix-caching pipelines through block-level sparsification and page-table remapping, enabling practical deployment while providing the memory savings needed for 128k+ long-context workloads.
>
> [1] Mooncake: A KVCache-centric Disaggregated Architecture for LLM Serving. https://arxiv.org/abs/2407.00079
> [2] ChunkKV: Semantic-Preserving KV Cache Compression for Efficient Long-Context LLM Inference. https://arxiv.org/abs/2502.00299
>
> ---
>
> > **`Q2: What is the computational cost of evaluating Eq. (7) in real-time decoding, and how is it amortized?`**
>
> Thank you for the question. In our reported experiments, ReST-KV is applied **only during prefill**, following the standard setup used by SnapKV and PyramidKV. During decoding, we reuse the compressed KV cache produced in prefill and do **not** evaluate Eq. (7) again, so there is no additional cost in the decoding phase for the results reported in the paper.
>
> If one wishes to extend ReST-KV to **decoding-time eviction**, Eq. (7) can be evaluated in a periodic manner (e.g., every $M$ generated tokens) on a small query window of size $w$. In this case, the incremental cost per eviction step is dominated by operations on the local window and the currently active keys, scaling as $O(wND)$ with a fixed, small $w \ll N$. Crucially, the attention statistics needed for Eq. (7) are already computed as part of the normal decoding pass, so the eviction step reuses these quantities and only performs a lightweight reconstruction-based scoring. Amortized over $M$ decoding steps, this results in a small per-token overhead while keeping the benefits of dynamic eviction, and the core complexity order remains aligned with attention-based methods such as SnapKV.
>
> ---
>
> > **`Q3: How would the approach behave under multi-turn interactive scenarios where KV eviction and reuse interleave dynamically?`**
>
> Thank you for the question. ReST-KV is compatible with the multi-turn interaction pattern used in modern LLM serving systems. Frameworks such as vLLM and SGLang process each turn by **concatenating the full dialogue history** into a new prompt, rather than maintaining a long-lived KV cache on GPU. This naturally triggers a new prefill step at every turn, during which ReST-KV re-computes KV importance within the latest query window. As a result, the method adapts cleanly to focus shifts across turns, even when eviction and reuse alternate over time.
>
> These systems also rely on **prefix caching** to reuse previously computed KV blocks. Under this workflow, ReST-KV simply reloads the needed uncompressed KV blocks from CPU, performs a lightweight query-window computation, and produces the compressed KV cache for the current turn.
>
> In summary, ReST-KV naturally supports multi-turn interactive scenarios: eviction occurs at each prefill step, reuse is handled through prefix caching, and the method remains robust even when the two processes interleave dynamically.
>
> ---

---

> > ### Comment · Reviewer_PCdR · 2025-11-26
> > **Thanks for response, I have increased the score.**
> >
> > Dear Auhors,
> >
> > Thank you for your response. I feel that my concerns have been satisfactorily addressed. I am raising my score from a 4 to a 6 and recommending the paper for acceptance at ICLR 2026.
> >
> > Bests,
> > Your Reviewer  PCdR

---

> > > ### Author Response · Authors · 2025-11-26
> > > **Acknowledgement of Reviewer PCdR’s Positive Feedback.**
> > >
> > > Thank you very much for your positive feedback and for raising your score. We are glad that our responses have satisfactorily addressed your concerns. We sincerely appreciate your careful evaluation and constructive suggestions, which have strengthened the overall quality of the paper. Thank you again for your time and for recommending our work for acceptance.

---

> ### Comment · Area_Chair_Xn2g · 2025-11-26
>
> Dear reviewer,
>
> Thanks for your time and effort in reviewing ICLR2026 submissions. The authors have submitted their responses to your review. Please take the time to read and raise your further comments, and discuss with the authors.
>
> Best regards,
>
> AC

---

### Official Review · Reviewer_4XR7 · 2025-11-03

**Soundness:** 3
**Presentation:** 2
**Contribution:** 2
**Rating:** 6
**Confidence:** 3

**Summary:**

This paper introduces a KV cache eviction method ReST-KV for LLM inference to improve robustness and efficiency under limited cache budgets. The key idea is to reformulate the eviction process as a layer-wise output reconstruction problem, estimating token importance by measuring output discrepancies when a KV pair is removed. The method further incorporates spatial-temporal smoothing through EMA and adaptive windowing to stabilize token selection over time. Experiments show improvements in accuracy and latency over prior eviction baselines.

**Strengths:**

- The authors clearly articulate the practical motivation behind efficient KV cache management.

- The proposed method is evaluated across multiple backbone models, and comprehensive ablation studies are conducted for each component.

- I have reviewed and reproduced the code in Appendix; it is well-organized and easy to follow.

**Weaknesses:**

- Measuring token importance by reconstruction error is essentially equivalent to attention-weight reweighting already explored in H2O, which models the same relationship between attention redistribution and output sensitivity.

- The EMA and adaptive window strategies are simple post-processing heuristics. Also, the authors do not include any hyperparameter sensitivity analysis.

- The evaluation focuses on retrieval-oriented long-context benchmarks, without including instruction-following or multi-turn dialogue scenarios.

**Questions:**

please see weakness

---

> ### Author Response · Authors · 2025-11-24
> **Official Comment by Authors**
>
> Thank you for the positive and constructive feedback. We appreciate your acknowledgement of the practical motivation for efficient KV management, the broad evaluation across backbone models, and the thoroughness of our ablations. We are also grateful for your comments on the clarity and reproducibility of the code in the Appendix.

---

> ### Author Response · Authors · 2025-11-24
> **[1/1] Response to Reviewer 4XR7 (W1~W4)**
>
> > **`W1: Measuring token importance by reconstruction error is essentially equivalent to attention-weight reweighting already explored in H2O.`**
>
> Thank you for the comment. We would like to clarify that the proposed indicator is fundamentally different from H2O. The H2O algorithm consists of two components: (1) retaining recent tokens based on temporal order (similar to sliding-window attention), and (2) selecting heavy hitters purely by **cumulative attention weights** over history. This accumulation-based scoring remains a function of $(q, k)$ only and does not model how the output would change if a KV pair were removed.
>
> In contrast, our formulation in Eq.(7) explicitly incorporates **value information**, making the indicator a function of $(q, k, v)$. This output-aware term approximates the leave-one-out change in MHA output and directly captures attention redistribution and output sensitivity—effects that attention-weight accumulation alone cannot represent.
>
> Therefore, the reconstruction-based metric is not equivalent to H2O’s reweighted attention score and provides strictly richer information for KV eviction. The substantial empirical gains across all benchmarks—including a +15.2% improvement on RULER—further support that modeling $(q,k,v)$-aware output change offers significantly stronger eviction decisions than attention-weight accumulation alone.
>
> ---
>
> > **`W2: The EMA and adaptive window strategies are simple post-processing heuristics.`**
>
> Thank you for the comment. Our method is designed to address an important aspect that prior KV eviction approaches largely overlook—**the effect of attention redistribution** when a KV pair is removed. To make the reconstruction-based indicator in Eq.(7) stable in a training-free setting, it is essential to model the **temporal variation** and **spatial drift** of KV importance observed in real attention trajectories. The proposed EMA and adaptive window mechanisms are therefore not ad-hoc heuristics, but lightweight components specifically designed to capture these dynamics directly from inference-time signals. Their contribution is consistently validated across models and cache budgets in our ablations.
>
> ---
>
> > **`W3: No hyperparameter sensitivity analysis is provided.`**
>
> Thank you for the comment. We would like to clarify that Appendix G.3 includes a dedicated sensitivity analysis (Figure 10) for both hyperparameters $\alpha$ and $\beta$. The results show that ReST-KV remains consistently better than the baselines across a *wide* range of values for both parameters. This robustness motivated our use of a **single fixed configuration** ($\alpha = 0.3$, $\beta = 2000$) for all models and tasks, without any per-model or per-dataset tuning.
>
> ---
>
> > **`W4: The evaluation focuses on retrieval-oriented long-context benchmarks, without including instruction-following or multi-turn dialogue scenarios.`**
>
> Thank you for raising this point. We would like to clarify that ReST-KV is fully compatible with multi-turn dialogue settings commonly used in real LLM deployments. In mainstream serving systems such as vLLM and SGLang, multi-turn interaction is executed by **concatenating the full dialogue history** into a new prompt for each turn, rather than keeping long-lived KV caches on GPU. This naturally triggers a new prefill step, during which ReST-KV re-evaluates KV importance within the latest query window, allowing it to adapt to focus shifts across turns.
>
> These systems also employ **prefix caching** to reuse previously computed KV blocks and reduce TTFT. Under this workflow, ReST-KV only needs to reload the uncompressed KV blocks from CPU, perform a lightweight query-window computation, and produce the compressed KV cache for the current turn. This makes ReST-KV directly compatible with existing multi-turn serving pipelines, as also discussed in the vLLM community (see <https://github.com/vllm-project/vllm/issues/12254#issuecomment-2643383112>).
>
> In summary, although our experiments focus on long-context understanding benchmarks, ReST-KV naturally extends to instruction-following and multi-turn scenarios under standard serving practices, and the strong results across diverse models and tasks indicate broad applicability.
>
> ---

---

> ### Comment · Area_Chair_Xn2g · 2025-11-26
>
> Dear reviewer,
>
> Thanks for your time and effort in reviewing ICLR2026 submissions. The authors have submitted their responses to your review. Please take the time to read and raise your further comments, and discuss with the authors.
>
> Best regards,
>
> AC

---

### Official Review · Reviewer_uohW · 2025-11-04

**Soundness:** 3
**Presentation:** 3
**Contribution:** 3
**Rating:** 6
**Confidence:** 4

**Summary:**

This paper proposes ReST-KV, a novel KV cache eviction method that reformulates token importance as a layer-wise output reconstruction problem, explicitly accounting for attention redistribution upon token removal. It further enhances robustness via spatial-temporal smoothing: exponential moving average (EMA) for temporal dynamics and an adaptive window mechanism for spatial shifts. Evaluated across LongBench, RULER, Needle-in-a-Haystack, and InfiniteBench, ReST-KV consistently outperforms SOTA methods—e.g., +2.58% on LongBench, +15.2% on RULER—and achieves a 10.61× decoding speedup at 128K context with minimal accuracy loss.

**Strengths:**

* Principled formulation: The reconstruction-based importance metric (Eq. 7) is theoretically grounded and captures attention redistribution, a key limitation of prior attention-weight-only methods.
* Strong empirical results: Consistent gains across diverse benchmarks, models (Llama2/3, Mistral, Qwen, Gemma), and cache budgets (64L–1024L).
* Robust design: Spatial-temporal smoothing significantly improves stability, as validated by ablation (Table 3).
* Practical impact: 10×+ speedup and 36% memory reduction enable real-world deployment of long-context LLMs.
Compatibility: Integrates seamlessly with budget allocation (PyramidKV, AdaKV) and prefill acceleration (FlexPrefill).

**Weaknesses:**

* Computational overhead: Computing reconstruction error requires per-token MHA re-evaluation during prefill, which may slow down the prefill phase (though decoding benefits).
* Hyperparameter tuning: EMA factor α and spatial scale β require tuning, though sensitivity analysis shows robustness.
* Limited theoretical analysis: While empirically sound, formal guarantees on reconstruction error vs. downstream task performance are missing.
* Focus on prefill-only eviction: No dynamic eviction during decoding, which may limit adaptability in streaming settings.

**Questions:**

* What is the prefill-time overhead of computing the reconstruction-based indicator compared to SnapKV or H2O?
* Could the reconstruction loss be approximated more efficiently (e.g., via Jacobian or first-order Taylor expansion)?
* How does ReST-KV perform in multi-turn dialogue where context evolves incrementally over many turns?
Is the method compatible with sparse attention patterns (e.g., Longformer, BigBird) during prefill?

---

> ### Author Response · Authors · 2025-11-24
> **Official Comment by Authors**
>
> Thank you for the constructive and thoughtful review. We appreciate your recognition of our principled reconstruction-based formulation, which explicitly models attention redistribution beyond traditional attention-weight heuristics. We are also grateful for your acknowledgement of the strong empirical results across models, budgets, and benchmarks, as well as the robustness contributed by our spatial–temporal smoothing design. Your comments on the practical impact—particularly the decoding speedup and memory reduction—and the compatibility with existing budget allocation and prefill acceleration strategies are highly encouraging.

---

> ### Author Response · Authors · 2025-11-24
> **[1/3] Response to Reviewer uohW (W1, W2, W4)**
>
> > **`W1: Computational overhead — computing reconstruction error may slow down the prefill phase.`**
>
> Thank you for the comment. We understand the concern. As clarified in L472–473, ReST-KV computes reconstruction error **only within a small query window**, so the dominant computational cost is **the same order** as attention-based methods like SnapKV. In particular, both methods are dominated by the same term in our implementation (supplementary material, `rest_kv/restkv_utils.py`, L144):
>
> $O(wND)$ for a fixed window $w \ll N$.
>
> Although ReST-KV computes output-aware terms, the implementation first aggregates statistics through `attn_weights_mean` and `attn_outputs_mean` (supplementary material, L181–186), which keeps the additional overhead small and does not change the overall asymptotic complexity.
>
> | Method             | Prefill Scope                     | Output Computed | Dominant Complexity |
> |--------------------|-----------------------------------|------------------|---------------------|
> | Full Cache         | All $N$ queries × $N$ keys         | Yes              | $O(N^2D)$           |
> | SnapKV             | Last $w$ queries × $N$ keys        | No               | $O(wND)$            |
> | **ReST-KV (Ours)** | Last $w$ queries × $N$ keys        | Yes              | **$O(wND)$**        |
>
> Empirically, Table 14 in the supplementary material shows that ReST-KV increases prefill TTFT by only **≈3%**.
> To provide further clarity, we additionally report SnapKV’s TTFT under the same 128k setting (newly added in the revision):
>
> | Method   | 128k TTFT (ms)    |
> |----------|---------|
> | Full     | 28230   |
> | SnapKV   | 28510   |
> | ReST-KV  | 29082   |
>
> These results confirm that ReST-KV introduces negligible prefill overhead while achieving substantially more reliable KV eviction.
>
> ---
>
> > **`W2: Hyperparameter tuning — EMA factor $\alpha$ and spatial scale $\beta$ require tuning.`**
>
> Thank you for the observation. As noted in Appendix G.3, Figure 10 shows that both $\alpha$ and $\beta$ remain consistently better than the baselines across a **very wide** range of values. Following this robustness analysis, all experiments in the paper use a **single fixed configuration** ($\alpha = 0.3$, $\beta = 2000$), as stated in Appendix B. The strong and consistent performance across all models and tasks demonstrates that this setting is robust in practice and avoids per-model or per-task tuning.
>
> ---
>
> > **`W4: Focus on prefill-only eviction — no dynamic eviction during decoding may limit adaptability in streaming settings.`**
>
> Thank you for pointing this out. We follow the common practice in SnapKV and PyramidKV and adopt a prefill-only eviction strategy to clearly isolate and evaluate the effectiveness of our method. Nevertheless, ReST-KV can be extended to **decoding-time eviction** with minimal modification. A simple implementation is to trigger eviction every $M$ generated tokens, compressing the cache from $B\ell + M$ back to $B\ell$. Since attention statistics in the local window can be recorded during decoding, the eviction step only needs to execute the lightweight importance computation in lines 186–208 of `rest_kv/restkv_utils.py`, ensuring real-time feasibility for streaming scenarios.
>
> ---

---

> ### Author Response · Authors · 2025-11-24
> **[2/3] Response to Reviewer uohW (W3)**
>
> > **`W3: Limited theoretical analysis — while empirically sound, formal guarantees on reconstruction error vs. downstream task performance are missing.`**
>
> Thank you for the comment. We agree that it is important to clarify how our **layer-wise reconstruction objective** relates to the **final prediction error**. Below we show that the final output deviation is **upper-bounded by a weighted sum of layer-wise reconstruction errors**, which directly justifies our design.
>
> ---
>
> ### 1. Notation and setup
>
> Consider a Transformer with $L$ layers. For an input $x$:
>
> - Original model:
>   $$
>   h^{(0)}(x) = x,\\quad
>   h^{(l)}(x) = F^{(l)}\\big(h^{(l-1)}(x)\\big),\\; l=1,\\dots,L,
>   $$
>   $$
>   f(x) = h^{(L)}(x).
>   $$
>
> - Model with KV eviction (ReST-KV):
>   $$
>   \\tilde h^{(0)}(x) = x,\\quad
>   \\tilde h^{(l)}(x) = \\tilde F^{(l)}\\big(\\tilde h^{(l-1)}(x)\\big),\\; l=1,\\dots,L,
>   $$
>   $$
>   \\tilde f(x) = \\tilde h^{(L)}(x).
>   $$
>
> We define:
>
> - Layer-wise output deviation:
>   $$
>   e_l(x) := \\big\\|h^{(l)}(x) - \\tilde h^{(l)}(x)\\big\\|.
>   $$
>
> - Layer-wise reconstruction error (directly caused by eviction at layer $l$):
>   $$
>   r_l(x) := \\big\\|F^{(l)}(h^{(l-1)}(x)) - \\tilde F^{(l)}(h^{(l-1)}(x))\\big\\|.
>   $$
>
> Our reconstruction indicator in Eq.(7) is precisely designed to approximate $r_l(x)$ for each candidate KV pair.
>
> ---
>
> ### 2. Final output deviation
>
> The final output deviation is just the deviation at the last layer:
> $$
> \\|f(x) - \\tilde f(x)\\|
> = \\big\\|h^{(L)}(x) - \\tilde h^{(L)}(x)\\big\\|
> = e_L(x).
> $$
>
> To understand how $e_L(x)$ depends on the layer-wise reconstruction errors $r_l(x)$, we analyze the error propagation layer by layer.
>
> ---
>
> ### 3. Error recursion at each layer
>
> For layer $l$, we write:
> $$
> \begin{aligned}
> h^{(l)}(x) - \tilde h^{(l)}(x)
> &= F^{(l)}\big(h^{(l-1)}(x)\big) - \tilde F^{(l)}\big(\tilde h^{(l-1)}(x)\big) \\
> &= \underbrace{F^{(l)}(h^{(l-1)}) - \tilde F^{(l)}(h^{(l-1)})}\_{\text{reconstruction error at layer } l} \\
> &\quad + \underbrace{\tilde F^{(l)}(h^{(l-1)}) - \tilde F^{(l)}(\tilde h^{(l-1)})}\_{\text{propagation of previous-layer error}},
> \end{aligned}
> $$
>
>
> Taking norms gives:
> $$
> e_l(x)
> = \\big\\|h^{(l)} - \\tilde h^{(l)}\\big\\|
> \\le r_l(x) + \\big\\|\\tilde F^{(l)}(h^{(l-1)}) - \\tilde F^{(l)}(\\tilde h^{(l-1)})\\big\\|.
> $$
>
> Assume the perturbed layer map $\\tilde F^{(l)}$ is $K_l$-Lipschitz in its input:
> $$
> \\big\\|\\tilde F^{(l)}(u) - \\tilde F^{(l)}(v)\\big\\| \\le K_l \\|u - v\\|,\\quad \\forall u,v.
> $$
>
> Then the propagation term is bounded as:
> $$
> \\big\\|\\tilde F^{(l)}(h^{(l-1)}) - \\tilde F^{(l)}(\\tilde h^{(l-1)})\\big\\|
> \\le K_l \\big\\|h^{(l-1)} - \\tilde h^{(l-1)}\\big\\|
> = K_l\\, e_{l-1}(x).
> $$
>
> Therefore, for each layer we have the **Lipschitz recursion**:
> $$
> \\boxed{
> e_l(x) \\,\\le\\, r_l(x) + K_l\\, e_{l-1}(x).
> }
> $$
>
> ---
>
> ### 4. Unrolling the recursion to bound the output layer
>
> We start from $e_0(x) = 0$ (the input is unchanged) and apply the recursion:
>
> - For $l=1$:
>   $$
>   e_1(x) \\le r_1(x) + K_1 e_0(x) = r_1(x).
>   $$
> - For $l=2$:
>   $$
>   e_2(x) \\le r_2(x) + K_2 e_1(x)
>   \\le r_2(x) + K_2 r_1(x).
>   $$
>
> Proceeding inductively:
> $$
> e_L(x) \\,\\le\\, \\sum_{l=1}^{L} r_l(x)\\,\\prod_{k=l+1}^{L} K_k.
> $$
>
> Define weights
> $$
> w_l := \\prod_{k=l+1}^{L} K_k.
> $$
>
> Then:
> $$
> e_L(x)
> = \\|f(x) - \\tilde f(x)\\|
> \\le
> \\sum_{l=1}^{L} w_l r_l(x).
> $$
>
> This yields the key conclusion:
>
> $$
> \\boxed{
> \\|f(x) - \\tilde f(x)\\|
> \\le
> \\sum_{l=1}^{L} w_l r_l(x),
> \\quad
> w_l = \\prod_{k=l+1}^{L} K_k.
> }
> $$
>
> In words: **the final output deviation is upper-bounded by a weighted sum of layer-wise reconstruction errors**.
>
> Since our reconstruction-based indicator in Eq.(7) approximates $r_l(x)$ and our greedy selection minimizes these errors under the cache budget, ReST-KV is effectively minimizing an explicit upper bound on the global prediction perturbation.
>
> ---
>
> ### 5. Relation to downstream performance (informal)
>
> Once the output deviation $\\|f(x) - \\tilde f(x)\\|$ is controlled, its impact on downstream performance follows standard intuition:
>
> - Common losses (e.g., cross-entropy) are Lipschitz in the model output, so small changes in $f(x)$ imply bounded changes in per-sample loss.
> - For LLMs, per-sample loss is strongly negatively correlated with task error rate; keeping the loss close to the full-KV model typically preserves overall accuracy.
> - KV eviction is sample-aware: the reconstruction error is computed using the actual attention trajectory of the current input, making the bound meaningful per example.
>
> Therefore, minimizing layer-wise reconstruction error not only controls the final output deviation but also serves as a principled surrogate for preserving downstream task performance, consistent with the empirical gains we observe across all benchmarks.
>
> ---

---

> ### Author Response · Authors · 2025-11-24
> **[3/3] Response to Reviewer uohW (Q1~Q4)**
>
> > **`Q1: What is the prefill-time overhead of computing the reconstruction-based indicator compared to SnapKV or H2O?`**
>
> Thank you for the question. In our implementation, the reconstruction-based indicator is evaluated only on a small query window of size $w \ll N$, and its dominant computational cost matches that of attention-based eviction methods such as SnapKV. As detailed in the supplementary material (`rest_kv/restkv_utils.py`, L144), both methods share the same $O(wND)$ complexity term, and the additional output-aware operations (L181–186) are lightweight and do not alter the asymptotic cost.
>
> Empirically, Table 14 reports that ReST-KV increases prefill TTFT by only **≈3%**, comparable to SnapKV:
>
> | Method   | 128k TTFT (ms) |
> |----------|----------------|
> | Full     | 28230          |
> | SnapKV   | 28510          |
> | ReST-KV  | 29082          |
>
> Regarding H2O, we note that its eviction strategy requires storing a **full cumulative attention matrix**, which grows linearly with sequence length. For long-context settings (≥64k), this leads to **significant memory pressure and OOM** in practice, preventing us from obtaining reliable TTFT numbers for direct comparison..
>
> Overall, the reconstruction-based indicator introduces only a minor prefill overhead relative to SnapKV, while remaining substantially more scalable than H2O in long-context scenarios.
>
> ---
>
> > **`Q2: Could the reconstruction loss be approximated more efficiently (e.g., via Jacobians or first-order Taylor expansion)?`**
>
> Thank you for the question. Our formulation already uses a lightweight first-order approximation of the leave-one-out MHA output (Eq.(5)–(7)), which avoids any Jacobian computation and reduces the reconstruction cost to the **same dominant complexity** as attention-based methods such as SnapKV. As detailed in the supplementary material (`rest_kv/restkv_utils.py`, L144), both ReST-KV and SnapKV are dominated by the same $O(wND)$ term, where $w$ is the small query window.
>
> Because this approximation is already the minimal-cost way to estimate output change while remaining training-free, additional Jacobian- or Hessian-based techniques would introduce significantly higher overhead without improving practical accuracy. Empirically, the current approximation yields consistent gains across all models and benchmarks, indicating that it is both efficient and sufficiently accurate for KV eviction.
>
> ---
>
> > **`Q3: How does ReST-KV perform in multi-turn dialogue where context evolves incrementally over many turns?`**
>
> Thank you for the question. ReST-KV is compatible with the multi-turn interaction pattern used in real LLM serving systems. Frameworks such as vLLM and SGLang handle multi-turn dialogue by **concatenating the full dialogue history** into a new prompt at each turn, rather than maintaining long-lived KV caches on GPU. This design naturally triggers a new prefill step, during which ReST-KV re-computes KV importance within the latest query window. As a result, the method adapts to evolving conversational focus across turns without requiring any additional mechanisms.
>
> These systems also employ **prefix caching** to avoid recomputing previously processed KV blocks. In this workflow, ReST-KV only needs to reload the uncompressed KV blocks from CPU (or a remote KV store), perform a lightweight query-window computation, and generate the compressed KV cache for the current turn. This approach keeps memory usage low while maintaining full compatibility with existing multi-turn serving pipelines, as also noted in the vLLM community (see <https://github.com/vllm-project/vllm/issues/12254#issuecomment-2643383112>).
>
> In summary, although our benchmarks focus on long-context understanding, ReST-KV naturally supports incremental multi-turn scenarios under standard serving practices, and its consistent performance across diverse models suggests its applicability extends beyond the evaluated settings.
>
> ---
>
> > **`Q4: Is the method compatible with sparse attention patterns (e.g., Longformer, BigBird) during prefill?`**
>
> Thank you for the question. Yes, ReST-KV is compatible with sparse attention architectures. In practice, one can first apply the sparse attention pattern of Longformer or BigBird to obtain the corresponding attention mask, and then run ReST-KV on top of the resulting attention scores.
>
> We would also like to note that the reconstruction-based indicator can itself yield a **sample-adaptive sparse pattern**, since it evaluates the importance of each KV pair directly through output reconstruction rather than relying on a fixed hand-designed sparsity layout. This often results in a more flexible and input-dependent sparsification compared with using a static sparse pattern alone.
>
> ---

---

> ### Comment · Area_Chair_Xn2g · 2025-11-26
>
> Dear reviewer,
>
> Thanks for your time and effort in reviewing ICLR2026 submissions. The authors have submitted their responses to your review. Please take the time to read and raise your further comments, and discuss with the authors.
>
> Best regards,
>
> AC

---

> > ### Comment · Reviewer_uohW · 2025-11-28
> >
> > I appreciate the authors’ detailed responses, I am maintaining my original positive score.

---

### Author Response · Authors · 2025-12-02
**Summary for Area Chair**

Dear Area Chair,

Thank you for taking the time to oversee the final evaluation of our submission. Due to the unexpected early closure of this year’s rebuttal phase, we provide this consolidated comment to support your efficient understanding of our work, its core contributions, and the full rebuttal process.

We appreciate the thoughtful evaluations contributed by you and all reviewers. During the rebuttal period, we engaged in detailed exchanges with each reviewer and provided the necessary theoretical analysis, complexity analysis, and additional experiments to address their concerns.

Below is a summary of our core contributions as well as our exchanges with each reviewer and the key improvements made to the paper.

---

### **[1/6] Summary of Key Contributions**

Our work proposes ReST-KV, the first KV-eviction framework that models the attention redistribution induced by removing a KV pair—an effect that prior approaches relying solely on raw attention weights largely overlook. ReST-KV further captures the spatial–temporal dynamics of KV-pair importance, enabling substantially more robust eviction decisions. With computational overhead comparable to existing methods, it achieves +2.58% on LongBench, +15.2% on RULER, and yields a 10.61× reduction in decoding latency at a 128k context length.

---

### **[2/6] Response to Reviewer uohW**

- **Core Concern:** Computational overhead; limited theoretical analysis; applicability beyond prefill-only eviction.

- **Our Response:**
  - We clarified that ReST-KV shares the same computational complexity as SnapKV and showed 128k TTFT results demonstrating comparable latency, with <3% overhead relative to full attention.
  - We provided theoretical analysis showing that the final output deviation is upper-bounded by a weighted sum of layer-wise reconstruction errors.
  - We noted that prefill-only eviction was used for fair comparison and ablation, while ReST-KV naturally extends to decoding-time eviction, for which we outlined a concrete approach.

- **Status:** The reviewer appreciated our clarifications and maintained a positive score.

---

### **[3/6] Response to Reviewer 4XR7**

- **Core Concern:** Similarity of motivation to H2O; heuristic nature of EMA and AWS; applicability in multi-turn dialogue.

- **Our Response:**
  - We clarified the fundamental distinction from H2O: cumulative attention weights and reconstruction error capture entirely different signals, and H2O lacks value-aware modeling.
  - We emphasized that our main contribution lies in modeling attention redistribution and spatial–temporal variation in KV importance; EMA and AWS are effective training-free mechanisms to operationalize these principles.
  - We explained that ReST-KV is fully compatible with multi-turn dialogue through prefix caching and KV-cache recomputation, avoiding focus-shift issues in practical settings.

---

### **[4/6] Response to Reviewer PCdR**

- **Core Concern:** Limited novelty; compatibility with prefix caching.

- **Our Response:**
  - We highlighted that the core contribution of our work lies in modeling attention redistribution and the spatial–temporal dynamics of KV importance—an approach fundamentally different from prior eviction methods. The substantial gains on multiple long-context benchmarks further validate this novelty.
  - We clarified that ReST-KV is fully compatible with prefix caching and provided a concrete integration strategy, along with references to relevant discussions in the vLLM community.

- **Status:** The reviewer indicated that their concerns were satisfactorily resolved, **raised the score from 4 to 6, increased the confidence from 4 to 5, and recommended acceptance.**

---

### **[5/6] Response to Reviewer fJbv**

- **Core Concern:** Feasibility of the greedy approximation; missing recent baselines.

- **Our Response:**
  - We provided supporting evidence from prior work and explained that the empirical distribution of attention weights makes the local linearity assumption more robust in practice.
  - We added comparisons with LaCache (ICML'25), showing that ReST-KV consistently outperforms it across all evaluated settings.

---

### **[6/6] Response to Reviewer iKEa**

- **Core Concern:** Effectiveness beyond 256k context; computational overhead.

- **Our Response:**
  - We pointed out that our appendix includes InfiniteBench results covering tasks beyond 128k, including a 2000k+ Zh.QA task, where ReST-KV consistently outperforms SnapKV, demonstrating robustness in ultra-long contexts.
  - We clarified that ReST-KV has the same computational complexity as prior eviction methods and provided concrete latency measurements to substantiate our theoretical analysis.

---

We hope these clarifications address the reviewer’s concerns and contribute to a positive reassessment of our work.

Best regards,

The Authors

---

### Meta-Review · Area_Chair_vVr2 · 2025-12-21

**Summary:**

This paper proposes ReST-KV, a KV cache eviction framework for long-context LLM inference that reframes token importance as a layer-wise output reconstruction problem, explicitly modeling attention redistribution effects that are largely ignored by prior eviction methods relying on raw attention weights. The method further introduces spatial–temporal smoothing to stabilize importance estimation in a training-free setting. Extensive experiments across multiple models, cache budgets, and long-context benchmarks demonstrate consistent accuracy improvements and substantial decoding speedups.

Overall, reviewers found the motivation well-grounded and the empirical evaluation thorough. The main points of contention centered on perceived novelty, system-level compatibility with modern serving pipelines, justification of approximations, and robustness at extreme context lengths. These issues were addressed to varying degrees during rebuttal.

**Reviewer Concerns:**

(1) Multiple reviewers questioned whether reconstruction-based importance estimation would incur significant prefill overhead. The authors clarified that the indicator is computed only on a small query window, shares the same dominant complexity as SnapKV, and empirically increases TTFT by only ~3% at 128k. This concern was satisfactorily resolved.

(2) Concerns about the lack of formal guarantees were mitigated by a rebuttal providing a clear layer-wise error propagation bound, linking reconstruction error to final output deviation. While not a full formal guarantee, this level of analysis is appropriate for an inference-time systems paper and was viewed as largely sufficient.

(3) Some reviewers initially viewed the method as incremental. The authors clarified the fundamental distinction between attention-only scoring and output-aware, value-dependent reconstruction, and provided strong empirical evidence (e.g., large gains on RULER). This concern was partially to largely resolved, though perceptions of novelty varied.


(4) One reviewer raised serious concerns about incompatibility with vLLM/SGLang-style prefix caching. The authors provided a detailed block-level integration strategy based on storage–compute separation and page-table remapping, which fully addressed this issue. This led to a score increase from 4 to 6 and an explicit recommendation for acceptance.

(5) Questions about the greedy approximation and adaptive window smoothing were addressed through references to established pruning literature, intuition about attention structure, and robustness analyses. These responses were generally accepted, though some reviewers still viewed these aspects as heuristic rather than theoretically deep.

(6) The strongest remaining concern came from a single reviewer who questioned robustness beyond 128k. The authors pointed to InfiniteBench results covering 200k–2M tokens, showing consistent improvements over SnapKV. While explicit 256k/512k latency breakdowns were not provided, the evidence substantially weakens the original concern. This point remains partially outstanding but not blocking, especially given corroborating evidence from other benchmarks.

**Reviewer Scores:**

Reviewer uohW: Maintained a positive score after rebuttal.

Reviewer 4XR7: Remained cautiously positive; no unresolved technical blockers.

Reviewer PCdR: Raised score from 4 to 6 and explicitly recommended acceptance after system-level clarifications.

Reviewer fJbv: Maintained a positive score; concerns addressed through added baselines and explanations.

Reviewer iKEa: Maintained a negative score, primarily due to concerns about extreme-length scalability and approximation assumptions, despite rebuttal clarifications.

Overall, the discussion shows one clear negative reviewer, one reviewer converted from reject to accept, and three consistently positive but cautious reviewers.

---

### Decision · Program_Chairs · 2026-01-26

Accept (Poster)